# Software Architecture for Autonomous and Coordinated Navigation of UAV Swarms in Forest and Urban Firefighting

Ángel Madridano [1,*], Abdulla Al-Kaff [1], Pablo Flores [2], David Martín [1] and Arturo de la Escalera [1]

1 Intelligent Systems Lab (LSI), Universidad Carlos III de Madrid, Avenida Universidad 30, 28911 Leganés, Madrid, Spain; akaff@ing.uc3m.es (A.A.-K.); dmgomez@ing.uc3m.es (D.M.); escalera@ing.uc3m.es (A.d.l.E.)
2 Drone Hopper S.L., Avenida Gregorio Peces-Barba, 28919 Leganés, Madrid, Spain; pablo.flores@drone-hopper.com
* Correspondence: amadrida@ing.uc3m.es

**Abstract:** Advances in the field of unmanned aerial vehicles (UAVs) have led to an exponential increase in their market, thanks to the development of innovative technological solutions aimed at a wide range of applications and services, such as emergencies and those related to fires. In addition, the expansion of this market has been accompanied by the birth and growth of the so-called UAV swarms. Currently, the expansion of these systems is due to their properties in terms of robustness, versatility, and efficiency. Along with these properties there is an aspect, which is still a field of study, such as autonomous and cooperative navigation of these swarms. In this paper we present an architecture that includes a set of complementary methods that allow the establishment of different control layers to enable the autonomous and cooperative navigation of a swarm of UAVs. Among the different layers, there are a global trajectory planner based on sampling, algorithms for obstacle detection and avoidance, and methods for autonomous decision making based on deep reinforcement learning. The paper shows satisfactory results for a line-of-sight based algorithm for global path planner trajectory smoothing in 2D and 3D. In addition, a novel method for autonomous navigation of UAVs based on deep reinforcement learning is shown, which has been tested in 2 different simulation environments with promising results about the use of these techniques to achieve autonomous navigation of UAVs.

**Keywords:** UAVs; swarm; autonomous; navigation; software architecture

## 1. Introduction

Recent advances, both in the field of Unmanned Aerial Vehicles (UAVs) and Multi-Robot Systems (MRS), have led to the expansion in the use of this type of system in civilian and military tasks [1,2]. This expansion has made UAV swarms, commonly known as drone swarms, a technological tool with high applicability in many areas, among which stand out those in which the times of action and response are a key aspect to their success. It is therefore emergency-related work where rapid, efficient and coordinated action can result in a high reduction of personal, social and economic damage. For all this, in recent years, research currents have emerged to improve the autonomous and coordinated action of drone swarms in different environments, whose achievement focuses on providing society with an effective tool capable of supporting, undertaking different tasks, and facilitating the performance of law enforcement when acting in emergency situations [3–5] such as natural disasters [6–9], search and rescue of people [10–13] or forest fires and urban fires [14–16].

Today, wildfires continue to be established as one of the main causes of the most devastating natural disasters [17,18]. Throughout 2019, fires such as those declared in the Brazilian Amazon with a burnt area of 6.52 million hectares, fires in Australia in which 1.3 million hectares burned and cost the lives of 6 people, those originating in California burning 31,000 hectares and forcing the eviction of more than 200,000 people and, finally,

although to a lesser extent, the fires of Canary Islands that caused the loss of almost 20,000 hectares of natural landscape, show that, to this day, the fight against fires remains a field in which it is necessary to introduce essential technological solutions that facilitate and improve the different tasks carried out in the four classic phases of firefighting such as prevention, detection, extinguishing and damage assessment.

Fires are considered to be one of the vital dangers to wildlife, the wild or urban environment, and one of the main factors seriously affecting the economies of countries. In addition, fire fighting requires a large number of people to carry out dangerous activities, which unfortunately cause a large number of victims each year. Forest fires affect 67 million hectares (ha) worldwide per year, approximately 1.7% of the earth's surface [17], and cost more than 2 billion euros per year, considering both fire fighting and economic damage [18]. The social and environmental costs include damage to human health and deaths (estimated at 340,000 premature deaths per year due to fires) [19] and significant damage to wildlife and soil, causing deforestation and release of greenhouse gases, linked to direct economic impact on the landscape (tourism) and damage to infrastructure, which are also considered important.

In many cases, these accidents are caused by the lack of real time information on the state of the fire or by the difficulty to reach in time the point of origin of the fire. According to the report on economic losses due to fires in the United States [20], there were about 1,298,000 fires in 2014, which caused 15,775 injuries and an estimated economic cost of $11.6 billion in direct property losses.

The possibility of being able to deploy in the same area two or more UAVs capable of working in a coordinated way is presented as a high added value solution in the field of forest fires because, not only can firefighting be carried out, but swarms can consist of UAVs of different characteristics and with different payment charges on board with which to undertake, simultaneously, different tasks such as monitoring a certain region of interest, monitoring of land equipment deployed in the area, capturing and processing essential fire information, generating and expanding communications networks or, accessing remote locations.

The set of properties described in the previous paragraph leads to different lines of research, such as this work, studying, analyzing and establishing the development and implementation of methods, encompassed within a software architecture, that allow these swarms to acquire, in addition, the possibility of navigating and undertaking work autonomously, without the need for human supervision deployed in the affected area, thus reducing the exposure of people to danger and reducing the possibility of personal harm. In this way, achieving the objectives of this work allows to have a highly powerful and effective tool such as a swarm of UAVs capable of navigating autonomously and coordinated through a multi-layered architecture that gives each agent sufficient intelligence for decision-making to allow it to establish safe navigation paths through the environment.

## 2. Proposed Architecture

The proposed software architecture consists of a set of layers oriented, each of them, to provide the swarm with the necessary technology to solve problems arising from coordinated and unsupervised navigation within the same environment (Figure 1). Each layer includes a set of methods that increase the robustness of the architecture, by developing redundant implementations based on different technologies, and allow to establish different control loops, at a high level, for the safe development of autonomous navigation of the UAV swarm.

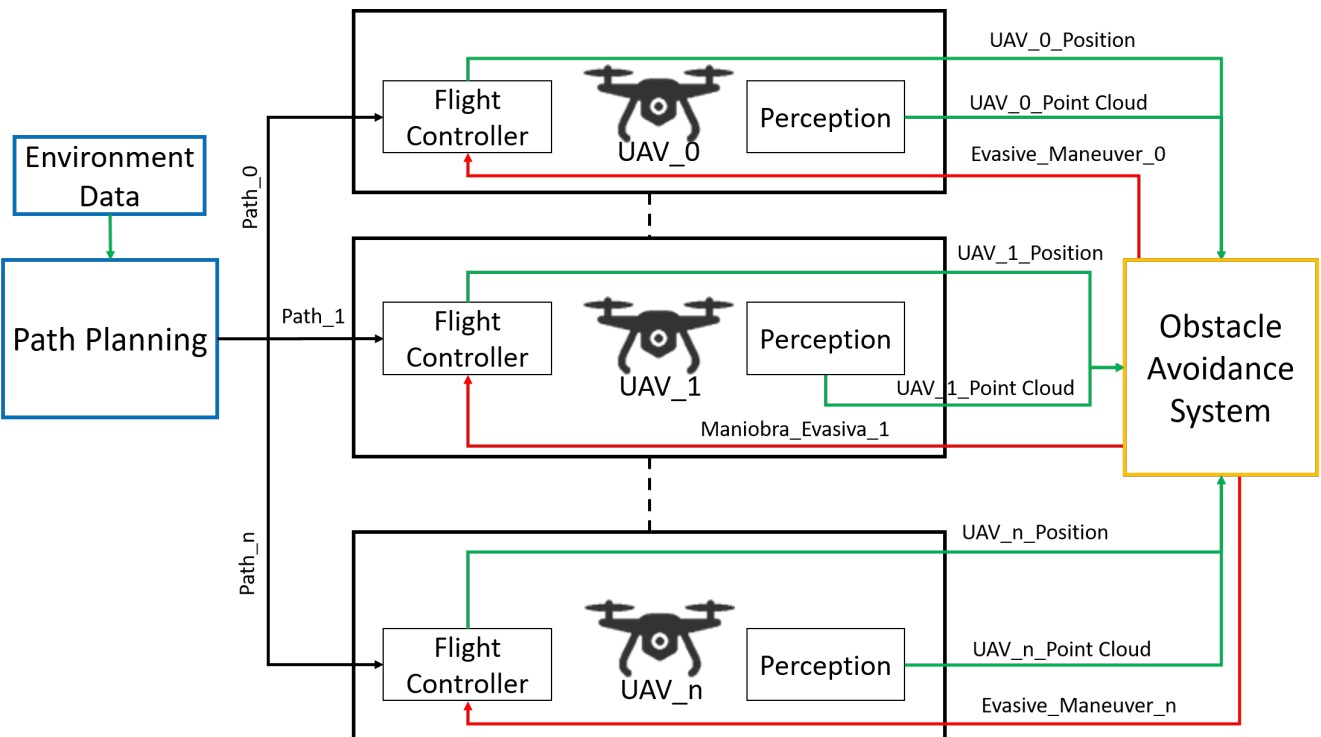

**Figure 1.** General scheme of the proposed autonomous Unmanned Aerial Vehicle (UAV) swarm.

The proposed architecture set is designed on the Robot Operating System (ROS) framework due to its ability to deploy and develop collaborative and portable software. Although these properties do not establish the only reasons why ROS has been used as the basis for the developed architecture, but that the need to test and validate each of the methods implemented on simulation, are another reason why the use of ROS is justified. And it is that, in the field of autonomous vehicles and, In particular, in the area of UAVs, simulations prior to implementation on real aerial platforms is a crucial step forward because, whatever error, however minimal, when testing each layer of architecture on a real UAV, can lead to loss or failure in the control of the UAV and, as a result, the complete loss of the vehicle, including the payment load shipped therein. For this reason, together with quantitative results, qualitative results derived from the simulation of each of the methods through the Gazebo simulator are analyzed, on which a swarm of UAVs is established with simulated models similar to real aerial platforms, in particular, in reference to the control system or autopilot, since both ROS and Gazebo allow to integrate the FIRMWARE of PX4 used by the Pixhawk controller, ensuring the correct implementation of the validated methods in simulation on a real swarm, as detailed in the work [21].

### 2.1. Layer I: Global Path Planner

The first layer of architecture is fully related to a key aspect within the autonomous movement of unmanned systems, such as path planning. The implemented path planner relies on both two-dimensional and three-dimensional information to establish, for each swarm agent, an optimal and secure path. To do this, from previous information of the environment, an exploration of the environment is carried out by using an algorithm based on Probabilistic Road Maps (PRM), which allows efficiently to establish, on an area of interest, a set of possible paths, free of collisions, on which to perform a safe autonomous navigation.

The use of this algorithm allows, against other methods of planning trajectories collected in the literature, to perform the exploration of large environments in a short period of time and, generate as an output a graph with all possible paths to be traveled. This unique and fast exploration of the environment allows you to establish a highly

scalable path planner for use in swarms of UAVs made up of a variable number of agents. Together with the PRM algorithm in charge of exploration, the A* algorithm is used to generate an optimal solution, in terms of total distance traveled, in such a way that, from the set of possible paths, establish, for each UAV, that path that allows it to reach a destination location traveling as little distance as possible [21,22].

This path planner not only has the advantage of being able to generate an optimal solution for a scalable number of UAVs or, being able to work with information in two or three dimensions of the environment, but also contemplates the possibility of being used for different situations such as: a labeled case, in which the location to which each swarm agent must go is previously known; an unlabeled case, for which the proposed path planner not only generates the paths, but is previously responsible for establishing which combination UAV target locations minimizes the total distance traveled by the swarm by using the Hungarian method; Finally, in order to be able to undertake, in the future, firefighting work optimally with a swarm of UAVs, the proposed path planner, allows to establish the final positions each swarm agent within a specific geometric formation and then establish the set of optimal and safe trajectories so that each UAV reaches its position within the formation.

While it is true that, the type and number of formations that a swarm of UAVs can achieve is very extensive and diversified, these three types of specific formations have been established in this work as the application of the swarm to work and work related to the fight against fire and possible tasks of extinguishing them have been established.

Forest firefighting is often linked to attacking and extinguishing combustion that usually spread uncontrollably over a plant and open area [23]. It is important to note that forest fires can be generated under a set of various geometric shapes, among which are some regular shapes such as circles or ellipses, and whose properties are determined by aspects such as weather conditions, orography, or variation in fuel type [24,25].

In the case of circular fires, the front of the fire is deployed in all directions from the central area to the outside [26], so the square formation, with a size and perimeter greater than the diameter of the circle of fire, would allow to act on the fire, avoiding its propagation and helping to control and reduce it (Figure 2).

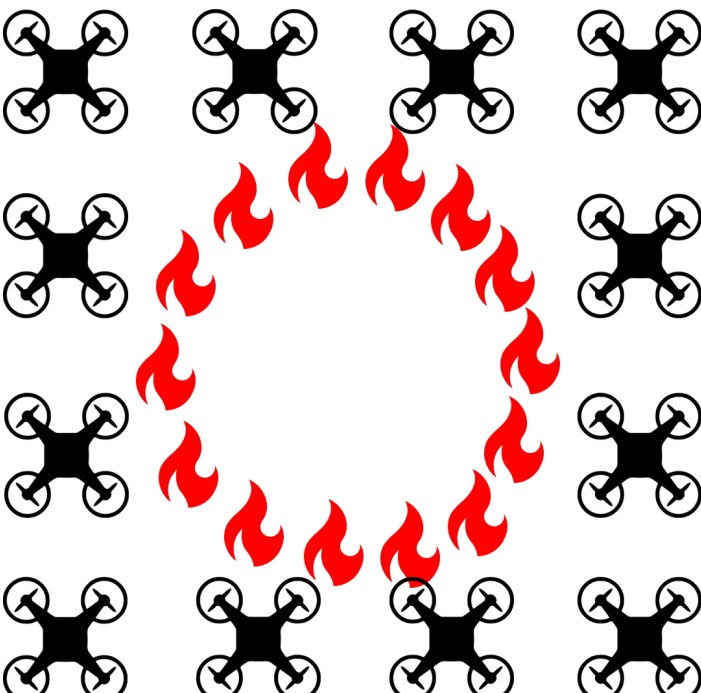

**Figure 2.** Action in square formation in front of a fire with a circular shape.

In the case of fires with an elliptical shape, they are characterized because the propagation is affected by a wind with a predominant direction [26]. In this case, when the fire advances towards a specific direction, having an arrow formation allows to act in an incisive way on the area of growth and advance of the fire. In this case, the size of the arrow should be established as a function of the smaller diameter of the ellipse, as shown in Figure 3, to act on the entire area affected.

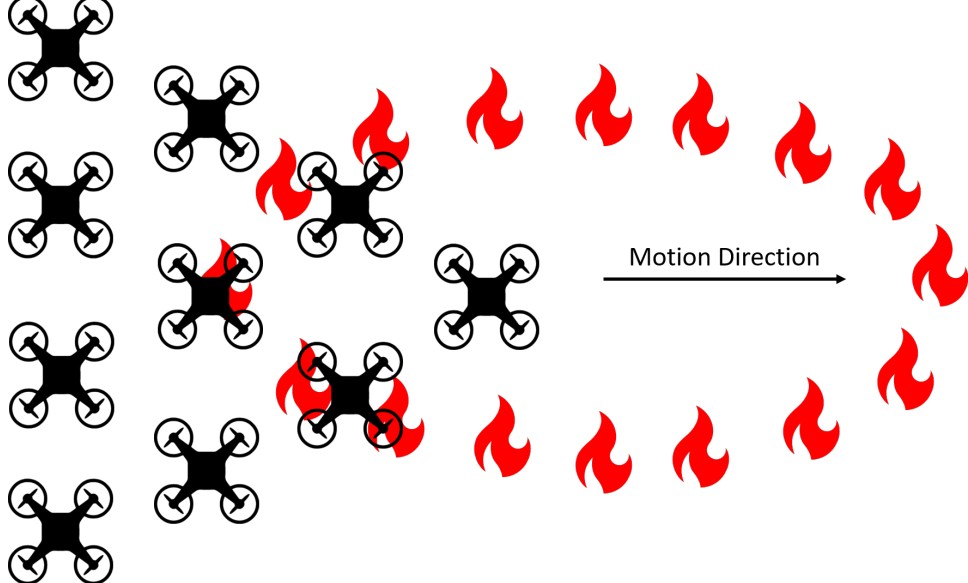

**Figure 3.** Action in arrow formation in front of a fire with an elliptical shape.

Finally, even if the fires do not have a regular shape, they do have different parts in common, among which the flame front stands out. This front is the perimeter part of the fire that is affected by wind speed and orography, so it is the area with the highest propagation speed and heat generation [26]. For this reason, it is considered the area of the fire that causes the greatest damage and, its containment is a crucial aspect for the control of the fire. An in line formation allows to act on that front effectively if the UAVs are provided in such a way that the length of the training line is like the length of the flame front (Figure 4).

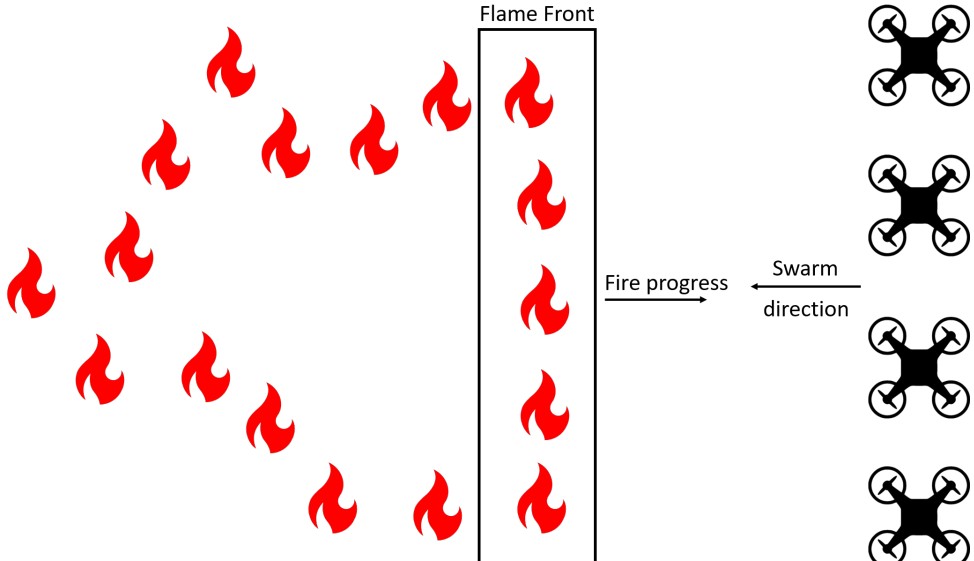

**Figure 4.** Action in line formation in front of a fire.

## *2.2. Layer II: Smoothing Trajectories*

Despite the advantages presented, path generation through PRM-based probabilistic exploration presents the drawback that the paths generated tend to present a Zig-Zag pattern that can affect the flight dynamics of UAVs [27,28]. For this reason, the implemented path planner combines, in a final phase, with a path smoothing algorithm responsible for removing unnecessary path points, thus improving the pattern of the paths, favoring the achievement of long stretches of straight lines and, in addition, optimizing the total distance traveled by the swarm and, with it, response times and efficiency when completing a mission.

PRM-based sampling algorithms have the disadvantage of generating paths with a Zig-Zag pattern, that is, achieving edges to establish a full path results in a broken path that goes against attenuated and constant dynamic behavior and, which generates abrupt changes in vehicle dynamics. This result, which is first associated with dynamic behavior not suitable for performing tasks such as capturing information through optical systems, or to carry out high-precision maneuvers and which, secondly, can cause premature maintenance of certain components of UAVs, is solved by a post-processing that allows to smooth the generated trajectories.

Among the solutions in the literature for correcting the problem of trajectories in Zig-Zag are the Line of Sight-based methods (LoS) that try to reduce the number of nodes or path points that make up the preset path. This post-processing allows to simplify the proposed path in a simple and computationally light way.

These algorithms consist that for each of the nodes entered in the generated path and, starting from the initial position of the UAV, it is checked whether there is a direct connection between that starting position and the target to be achieved. This plots a segment between the two points and checks whether that segment is collision-free. If this situation does not occur, an iterative process is performed in which the join segment between the start point and each of the nodes in the default path is checked, but in reverse, that is, from the destination path point to the initial node. If an obstacle-free segment is found, this border is set to good and all intermediate edges and nodes that were part of the preset path are removed. The entire process is then repeated, but instead of searching for a segment that joins the start point with the end, the intermediate node already connected to the starting position is taken as a starting point, and the obstacle-free segment search process is repeated from the target location and backwards through the established path points until the collision-free segment is found.

The algorithm developed has been applied to both 2D planning solutions and solutions generated in a 3D environment. It is important to note that, in the case of smoothing paths over 3D, the computation time used by the algorithm is superior to the need to work with large amounts of information such as 3D occupancy maps. The path smoothing process requires an iterative process, in the for each path, you must check for collision-free segments between nonconsecutive nodes to remove intermediate edges and nodes, implementing a procedure like that used to generate the edges of the 3D graph.

1. **Smoothing paths for 2D planning.**

   By analyzing the results for 2D exploration and planning, you can see how using slightly longer computation time, reducing the number of path points entered in the paths, and thereby reducing by a small proportion the total distance traveled by the swarm.
   Figure 5 shows the qualitative result of this algorithm for a single path, so that it can be observed more clearly how the algorithm eliminates those unnecessary intermediate nodes, since it is possible to connect one node to another further away without going through that intermediate point. Figure 5a shows on a 2D plane what the original path is, while Figure 5b shows the process of removing intermediate nodes and thus a path that reduces the Zig-Zag effect discussed above.

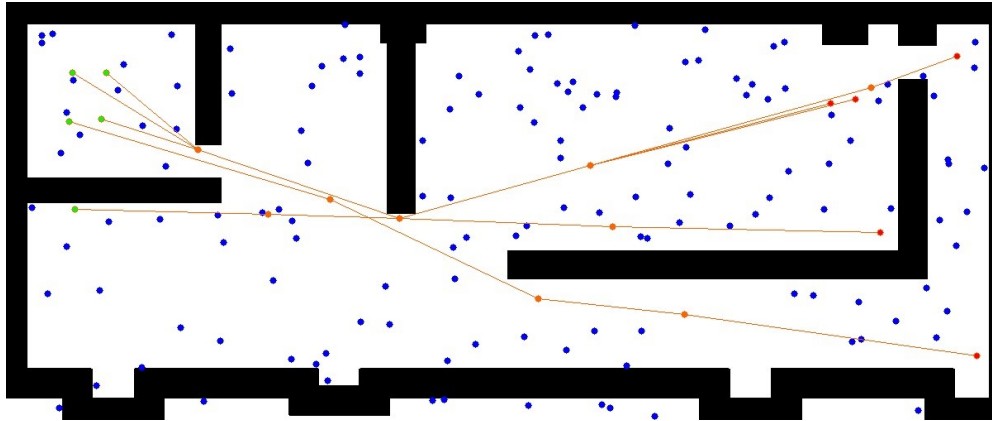

(**a**) Trajectory planning for Original 2D case.

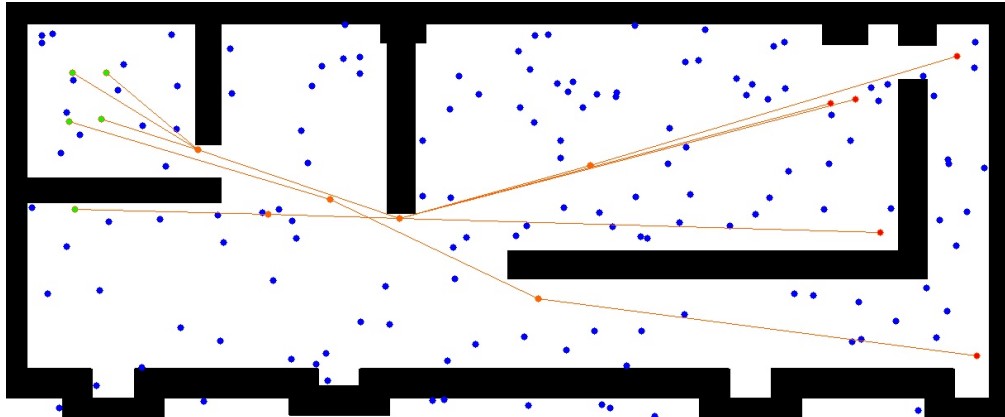

(**b**) 2D map with smoothed trajectory.

**Figure 5.** Example of 2D path smoothing.

Analyzing the algorithm from a quantitative point of view, the number of path points generated in the whole of the paths, the computational time derived from the implementation of that method or the total distance traveled by the swarm depending on the type of path used are aspects to deal with. To this end, it has been decided to compare the solution generated by the 2D path planner for both situations, considering only the labeled case, that is, in which the objective assignment-UAV is correlative to the order of registration of the aircraft and the final locations. The experiments have been conducted considering a swarm of 5 UAVs and, since the exploration of the environment is random, 5 different attempts have been made on the same scenario to be able to analyze the data more accurately.

The first aspect to consider is the number of path points generated in each of the paths and how with the use of the smoothing algorithm it is possible to reduce, in most cases, the number of path points involved in each of the paths of the UAVs, as shown in Table 1.

**Table 1.** Comparison of waypoints entered in the original and smoothed 2D paths for each UAV in the swarm.

|  | Test 1 | | Test 2 | | Test 3 | | Test 4 | | Test 5 | |
|---|---|---|---|---|---|---|---|---|---|---|
| **Paths** | **O** | **S** | **O** | **S** | **O** | **S** | **O** | **S** | **O** | **S** |
| $UAV_1$ | 5 | 4 | 5 | 3 | 6 | 4 | 6 | 4 | 5 | 4 |
| $UAV_2$ | 6 | 4 | 6 | 5 | 6 | 5 | 5 | 4 | 6 | 5 |
| $UAV_3$ | 6 | 4 | 7 | 5 | 6 | 5 | 5 | 5 | 5 | 5 |
| $UAV_4$ | 5 | 4 | 6 | 5 | 6 | 5 | 5 | 4 | 6 | 5 |
| $UAV_5$ | 4 | 4 | 4 | 3 | 5 | 4 | 4 | 4 | 4 | 4 |

O = Original Path, S = Smooth Path.

Although individually the reduction of path points may seem small, if you analyze their impact on all the paths generated for swarm it is observed that the overall is greater, as shown in Figure 6. It is drawn from the data collected in that figure that, in some cases, the reduction of the path points is greater than 20% and that, the mean is very close to it, with a value of 19.41%, which for a small scenario such as Figure 5 is not a significant number of path points, but that for large environments, a 20% reduction in the intermediate nodes to be achieved can result in a significant improvement in both the dynamic behavior and the total time spent by the UAV in covering that trajectory. If, in addition, one of the path points is present in the path of each UAV, which will always be present, and the reduction of path points is analyzed considering only the intermediate nodes, the reduction percentage rises to 23.85% of all possible nodes to be deleted, since that path point corresponding to the destination is part of all paths.

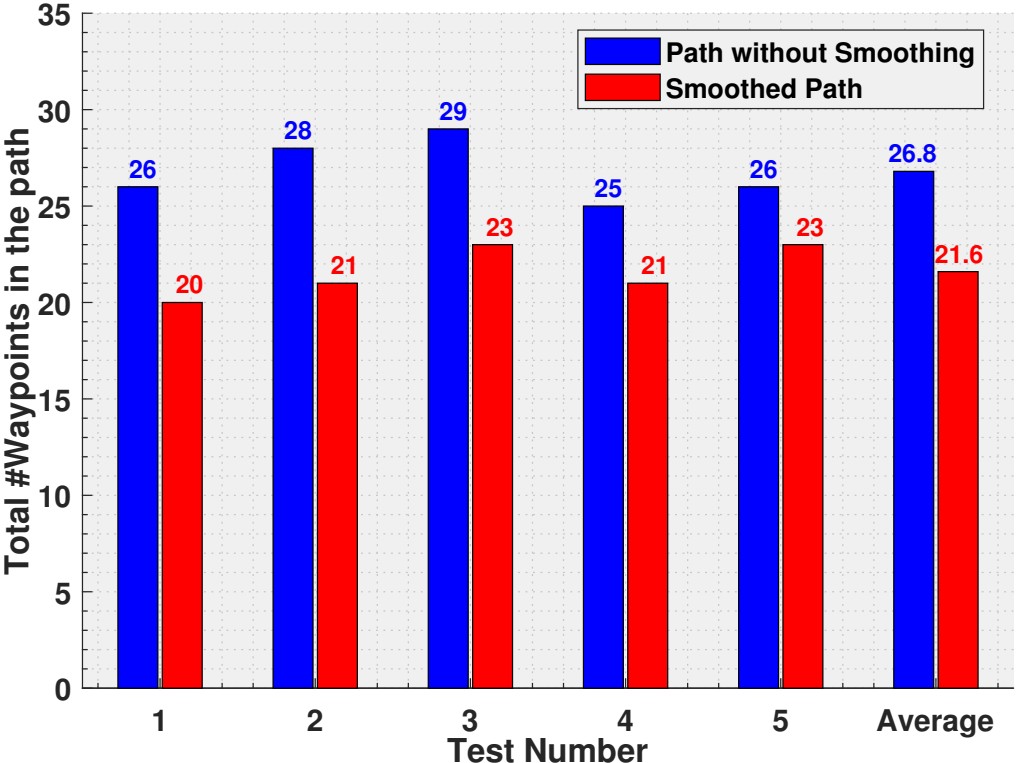

**Figure 6.** Comparison of total waypoints for original paths and 2D smoothed paths for a swarm of 5 UAVs.

Another aspect to be analyzed is whether this reduction in path points results in a reduction in the total distance traveled by the swarm and that, assuming stable dynamic behavior in which UAVs navigate at a virtually constant speed, implies a reduction in response time. In this case, for the proposed 2D planning, the reduction is virtually negligible, as shown in Figure 7 for all experiments.

This is due to several reasons: the first is that the total travelled distance in Figure 7 considers only the distances and movement in the *XY* plane and therefore does not consider aspects such as the distance traveled during the take-off phase, nor possible variations in height, constituting this aspect as the key to this lack of distance reduction; and, secondly, the density of the constructed graph and the parameters of creating it are well adjusted since the number of path points originally entered is not too high, so the initial paths generated are approaching or are the optimal solution in terms of minimum distance traveled and dynamic behavior.

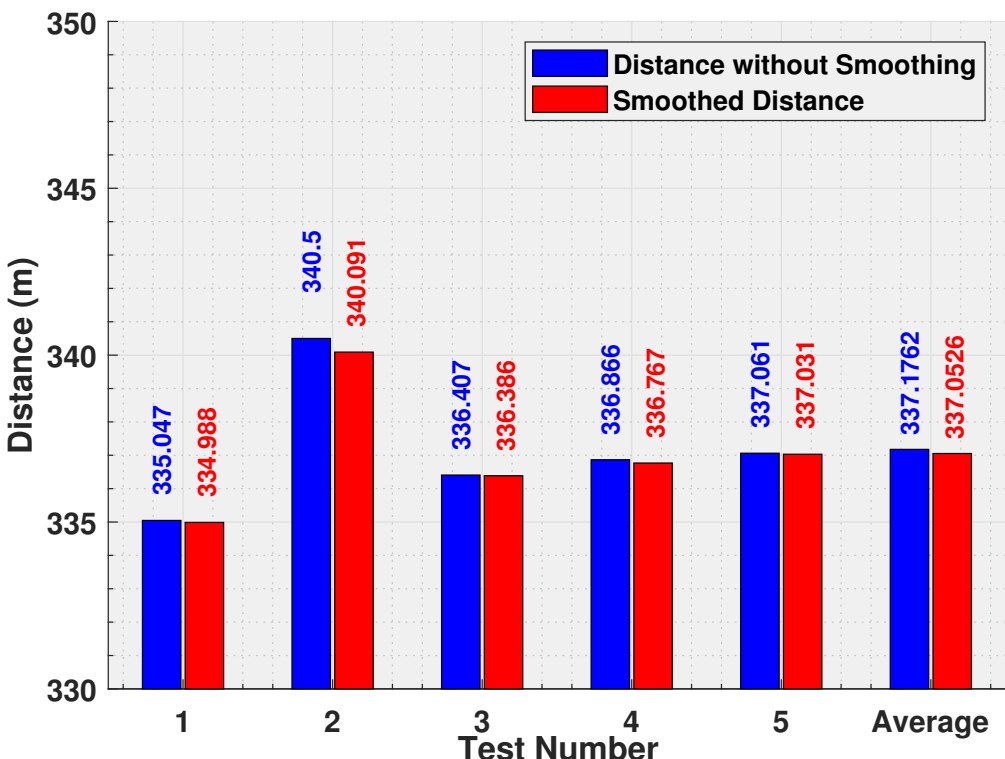

**Figure 7.** Comparison of the total travelled distance by a swarm of 5 UAVs for original trajectories and smoothed trajectories in 2D.

Finally, the other parameter analyzed with respect to the implementation of this algorithm is the additional computation time generated by its use to obtain an optimal solution, not only in distance traveled but in dynamic system behavior.

For the cases analyzed, it is noted in Figure 8 that the use of this algorithm leads to an increase in computational time of 6.62% on average, which leads to the conclusion that, given the reduction of the percentage of total paths points and, with it, the dynamic improvement of swarm movement and the possible reduction of navigation time towards the target, this algorithm could be employed at a post-processing stage to obtain a more optimal and complete solution at the cost of a slight increase in computing time spent obtaining such a solution.

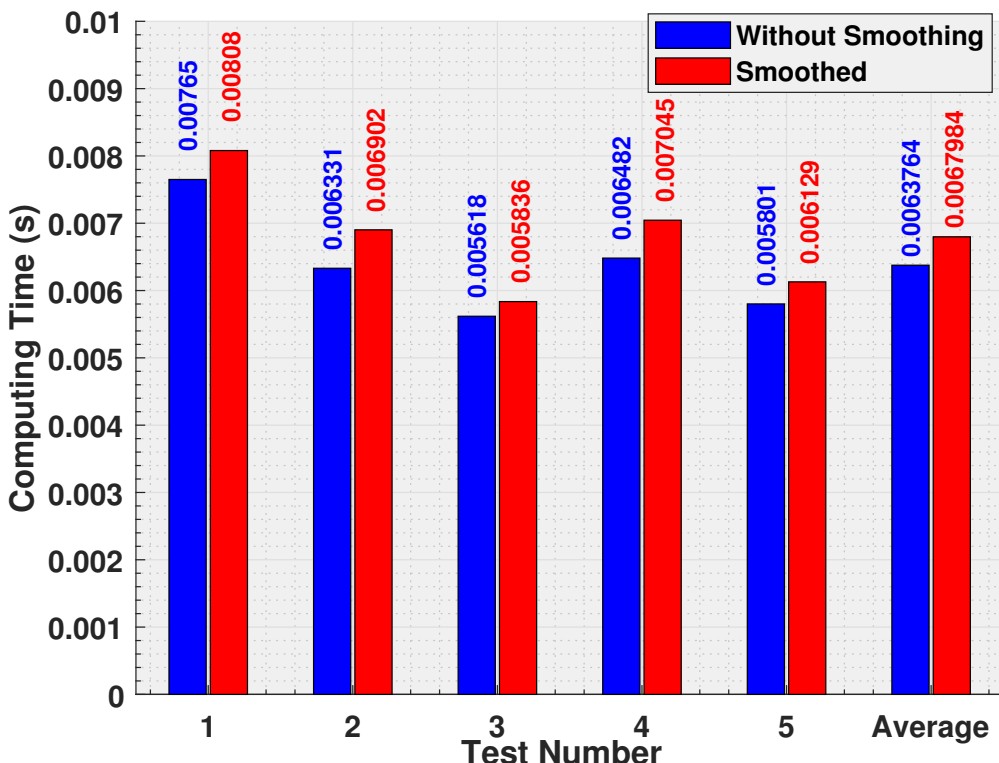

**Figure 8.** Comparison of the computation time used for the generation of original and smoothed trajectories in 2D for a swarm of 5 UAVs.

2. **Smoothing paths for 3D planning.**

As for the application of the smoothing algorithm to 3D paths, by including the Z coordinate and working on as much space as possible, the qualitative result of the algorithm is more evident, being the smoothing of paths more ostensible in 3D paths than in 2D paths, as can be seen when comparing Figure 9a, which contains the original solution generated for a swarm of 5 UAVs, considering only the distance traveled as a parameter to optimize, with Figure 9b, which shows the smoothed solution of those paths, adding to the distance optimization a better dynamic behavior of the system. The comparison shows how the change in trajectories is quite significant, eliminating the pattern in Zig-Zag and, generating trajectories in which long paths predominate in a straight line, thus improving, too, the response times of swarm performance.

As in 2D planning, a quantitative analysis of the use of this algorithm is collected as smoothing paths with Zig-Zag pattern analyzing their impact in aspects such as the number of path points included in the final solution, the total distance traveled by the swarm or the computation time required to generate trajectories that improve the dynamic behavior of the different UAVs.

As for the number of path points that are part of each of the paths, in the case in 3D it can be observed, as shown in Table 2, that for each of the participating UAVs the reduction carried out is considerable, which highlights the need for post-processing algorithms for the elimination of intermediate nodes.

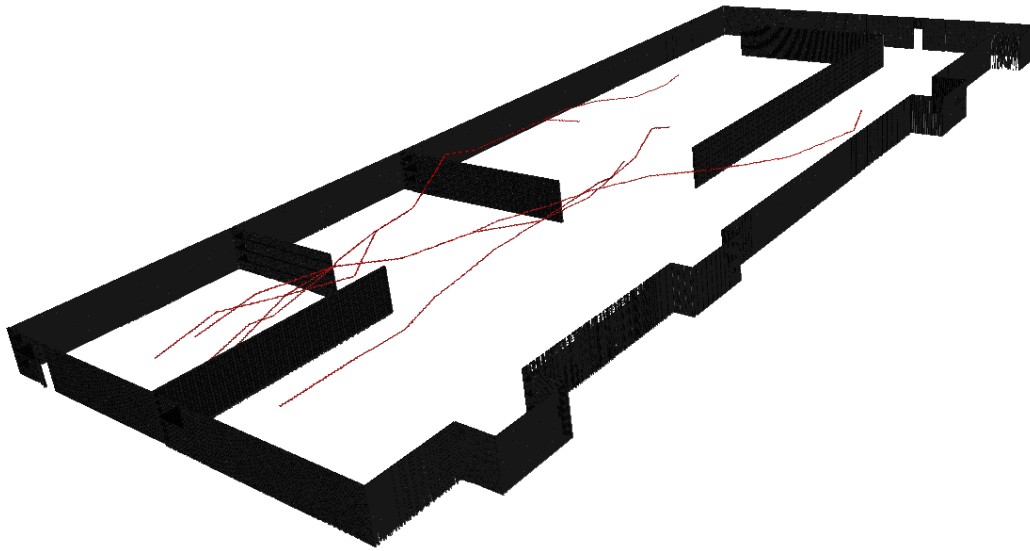

(**a**) Trajectory planning for original 3D case.

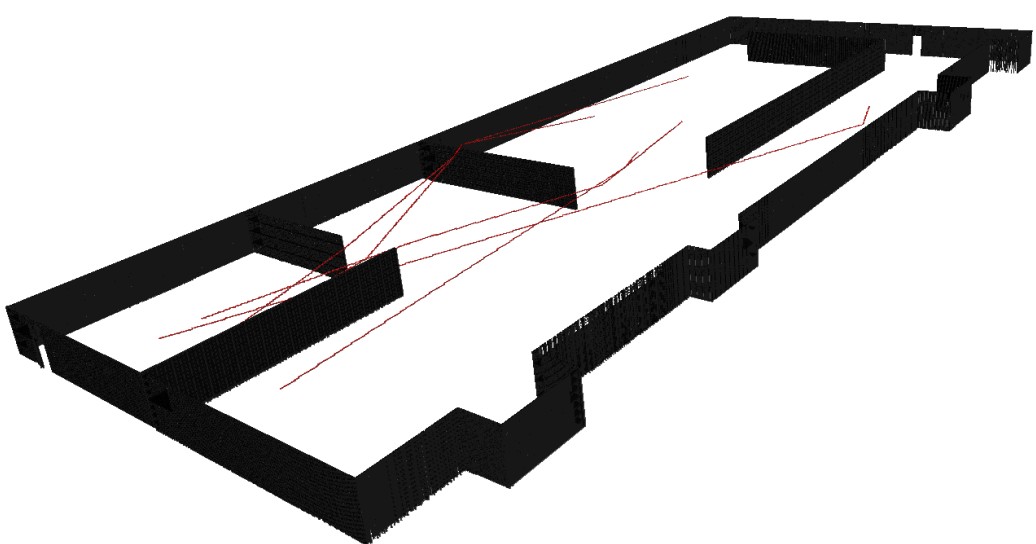

(**b**) 3D map with smoothed trajectory.

**Figure 9.** Example of 3D path smoothing.

**Table 2.** Comparison of waypoints entered in the original paths and smoothed in 3D for each swarm agent.

|  | Test 1 | | Test 2 | | Test 3 | | Test 4 | | Test 5 | |
|---|---|---|---|---|---|---|---|---|---|---|
| **Paths** | **O** | **S** | **O** | **S** | **O** | **S** | **O** | **S** | **O** | **S** |
| $UAV_1$ | 15 | 3 | 15 | 4 | 15 | 3 | 15 | 3 | 16 | 3 |
| $UAV_2$ | 13 | 3 | 13 | 3 | 13 | 3 | 13 | 3 | 13 | 3 |
| $UAV_3$ | 15 | 4 | 15 | 3 | 15 | 3 | 15 | 3 | 14 | 4 |
| $UAV_4$ | 12 | 3 | 11 | 4 | 12 | 4 | 13 | 3 | 13 | 3 |
| $UAV_5$ | 11 | 3 | 11 | 3 | 10 | 3 | 11 | 4 | 10 | 3 |

O = Original Path, S = Smooth Path.

Individually the reduction of path points is considerable, but if you analyze their impact on the set of paths created for swarm, it is observed that the overall impact is very significant, as shown in Figure 10, from which it can be extracted that in some

cases the reduction reaches 76% of the path points and, that the average is 75.38% of nodes eliminated, which undoubtedly represents a significant improvement both from the point of view of the dynamic behavior of UAVs, and the total time spent in navigation, since, when following the default paths, the speed of the UAV is reduced by approaching a path point in order to gain precision in the passage through it. In addition, as in the 2D case, it should be considered that from the set of nodes that make up each of the paths, one corresponds to the location of the target and, this node can never be deleted. Therefore, by bypassing this node and considering the percentage of reduction on which they can be eliminated, it is obtained that with the proposed algorithm a reduction of intermediate path points of 81.57% is achieved.

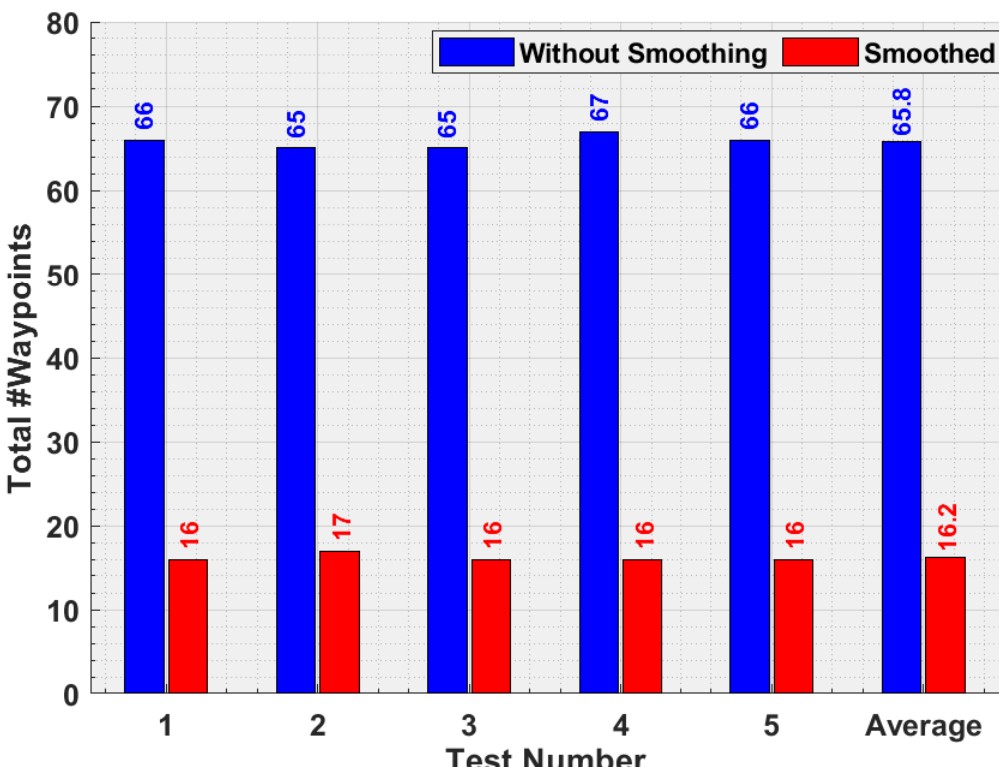

**Figure 10.** Comparison of total waypoints for original paths and 3D smoothed paths for a swarm of 5 UAVs.

The next aspect to be analyzed is how the reduction of intermediate nodes affects or does not affect the reduction of the total distance traveled by the set of UAVs and thus also a reduction in the time spent reaching the destination. Unlike the 2D case, in this case the reduction is somewhat more palpable, as shown in Figure 11, mainly due to the total consideration of the environment and not only of the X-Y plane and, also, because when planning on one more dimension, the generated graph is not as well adjusted as in the previous case, so the effects of smoothing are more evident. Although this reduction is in all cases above 5 m, its percentage value is low, located on an average of 3%, for a swarm of 5 UAVs.

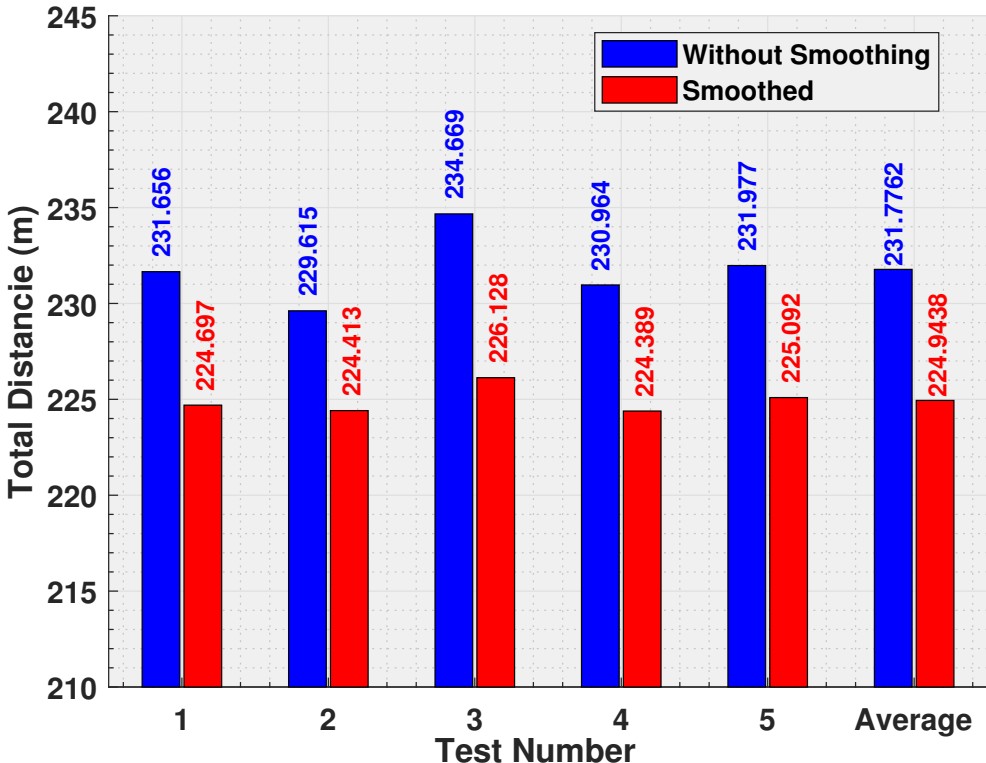

**Figure 11.** Comparison of the total distance traveled by a swarm of 5 UAVs for original paths and 3D smoothed paths.

Finally, it is essential to analyze how the use of this algorithm affects the computation time used to generate such an optimal solution, which improves the dynamic behavior of UAVs and, it is at this point that the proposed algorithm presents its main drawback, as shown in Figure 12. By using this algorithm over a large 3D environment, the time spent traversing the original paths, analyzing which nodes can be removed, and proposing a smoothed solution increases to more than 1 s. This considerable increase in computation time is due to the need to work with 3D occupation maps, which have useful and detailed information about the environment, but which cause the use and repeated access to such information to slow down algorithms such as the one proposed in this section. Unlike the proposed path planning algorithm, in which the occupancy map is processed only once to generate the network, in the case of the path smoothing method, three-dimensional information from the occupancy map must be accessed each time you want to check whether the segment joining two nodes is free of obstacles or not.

This result eliminates the use of this algorithm for applications that require dynamic calculation and smoothing of paths in real time but does not limit their use for a post-processing stage that allows to generate an optimal solution with better dynamic response of UAVs by removing the Zig-Zag pattern from the original paths obtained from the global solution built by PRM-based 3D path planning algorithms. Alongside this reason, it is important to consider that path smoothing and, therefore, the elimination of intermediate path points improves swarm response times and generates a more efficient solution in terms of distance traveled and mission execution time.

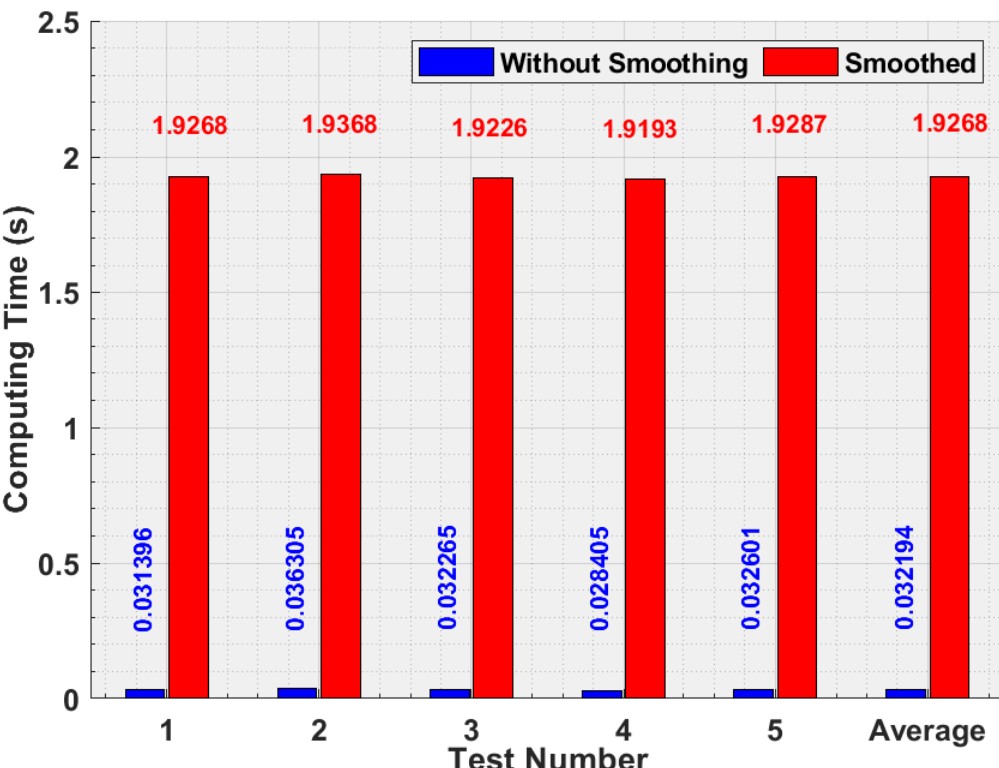

**Figure 12.** Comparison of the computation time used for the generation of original and smoothed trajectories in 3D for a swarm of 5 UAVs.

### 2.3. Layer III: Collision Detection and Avoid

The third layer of the developed software architecture ensures, on the one hand, coordinated navigation without swarm supervision and, on the other hand, collision-free navigation with objects present in the work space. Thus, this layer combines a set of methods that guarantee the free movement of UAVs through the paths established by the global planner or, through alternative paths generated in case the presence of new or dynamic obstacles prevent navigation through the previously determined paths [29].

The first of the methods established within this layer is based on a centralized control responsible for capturing the GPS position of each of the swarm agents, establishing the relative location of each other and determining different cruising speed profiles for each UAV in the event of a collision between two or more swarm agents. Thus, through the common knowledge of the positioning of each of the UAVs and, after establishing a safety margin, from which the speed control intervenes, that method is responsible for applying different speed reductions to each of the swarm agents if they approach the same location in the same time and, by nature, the possibility of collision between those drones is given. This cruise speed control allows each agent of the conflict to traverse the location in question at a different time and thus avoid collision between agents, allowing coordinated autonomous navigation of the system. In relation to this method, this second layer of architecture includes an implementation that seeks to avoid collisions in situations where, at least one of the UAVs, is taking advantage of multi-rounders to be able to stay on a stationary flight, without the need to make forward movements to generate support, as is the case with fixed-wing aircraft. In this case, to maintain the position on the *XY* plane of the UAV in question, the collision avoidance system is responsible, in detecting that event, for setting a different altitude value for each swarm agent involved in the conflict, thus avoiding possible collision [29].

The second of the methods is also related to collision detection and avoid, but in this case based on the capture and processing of three-dimensional information from the environment. Using sensors capable of capturing information in 3 dimensions of the

environment, each UAV, in a decentralized manner, can detect the presence of obstacles within the fixed paths to follow and, if necessary, begin a process of searching and establishing alternative paths that allow to reach a location of interest avoiding collisions with objects present in the environment. This second decentralized method allows to generate robustness when navigating without supervision detecting and avoiding collisions, since, in the face of the loss of any communication, the previous method may not detect the proximity between two UAVs, but through the 3D information captured by each agent, the drones involved may be aware of such collision and, each of them establish an alternative path that avoids such collision. Therefore, the combination of these 2 methods allows this second layer to establish control over the different swarm agents to ensure safe and coordinated navigation through previously set paths or, through secure alternative paths that avoid collisions between swarm UAVs or objects not collected in previous environment information [29].

Therefore, in this layer of the architecture, a set of methods oriented to collision detection and avoidance are established. Both methods are complementary and generate robustness in the architecture. Since the autopilot can receive control commands in both position and velocity, but not both simultaneously, the idea is that both methods generate as output a velocity control, so that both signals can be complementary. This is possible because the first algorithm, in charge of modifying the velocity profiles in case of approach between swarm agents, acts on the velocity modulus and not on the direction vector. While the second algorithm, capable of establishing a safe alternative path, oversees generating the direction of the velocity vector. Therefore, both systems are redundant and can act simultaneously avoiding collisions with both swarm agents and external obstacles.

The development on ROS allows to establish this high modularity within the architecture. In this way, although all navigation is established on a centralized architecture, ROS allows each UAV to have an individual node where the information captured by the sensors is processed and a velocity direction vector is generated as a response. This response can be introduced in the information coming from the centralized method, introducing to the autopilot a single velocity control command formed by a vector with a specific modulus and direction. In this way, it is possible to maintain a centralized swarm architecture and at the same time have, in each UAV, a decentralized control node that allows navigation to continue in case connectivity between the UAV and the central node on the ground is lost.

The high versatility of UAV swarms causes each agent to develop a different task on certain occasions. Among the different tasks are those related to the monitoring and monitoring of a particular area that require the maintenance of a UAV at a certain height without varying that position in the *XY* plane. This situation of staticity or hover causes that, collision avoidance systems between UAVs of the same swarm as described in the previous work and based on cruise speed control, are not useful for UAVs in hover maneuvers, so it is necessary to contemplate other solutions that allow UAVs working on these tasks, perform avoidance maneuvers without losing the objective of their mission. This static situation also joins the possible situation that some of the UAVs do not have the on-board technology needed to implement the environment perception-based obstacle detection method.

For this reason, a new method is presented, based on the traffic alert and collision avoidance systems used in manned commercial aircraft and, which is responsible for anticipating possible collisions between the different aircraft, working independently of air traffic services. Its operation is based on an on board transponder system that is responsible for exchanging distance, course and altitude information, with similar systems shipped on other nearby aircraft. According to the standard established by International Civil Aviation Organization (ICAO), there are three possible anti-collision systems, classified according to their performance, ranging from a standard I, in which only one alert is provided, to the standard III, still in development, in which an alert is provided along with 3D conflict resolution. In the intermediate position, there is standard II, which resembles the proposed method, and is based on a traffic alert plus a resolution of conflicts in the vertical plane [30].

For this reason, a collision avoidance method for UAV swarms based on virtual perimeters is presented around each of the swarm agents. The development of UAV applications related to these virtual geographic barriers have increased in recent years focused, above all, on the future integration of UAVs together with traditional aviation into a shared airspace [31–33]. Such a method would provide the proposed architecture with a new redundant layer for collision avoidance between swarm UAVs.

In this case, a method is established over the implemented swarm architecture that allows hover UAVs to perform maneuvers involving vertical movements to avoid possible collisions with other swarm UAVs that navigate unsupervised within a nearby radius. To do this, two safety areas of different spokes are established around each agent, as shown in Figure 13.

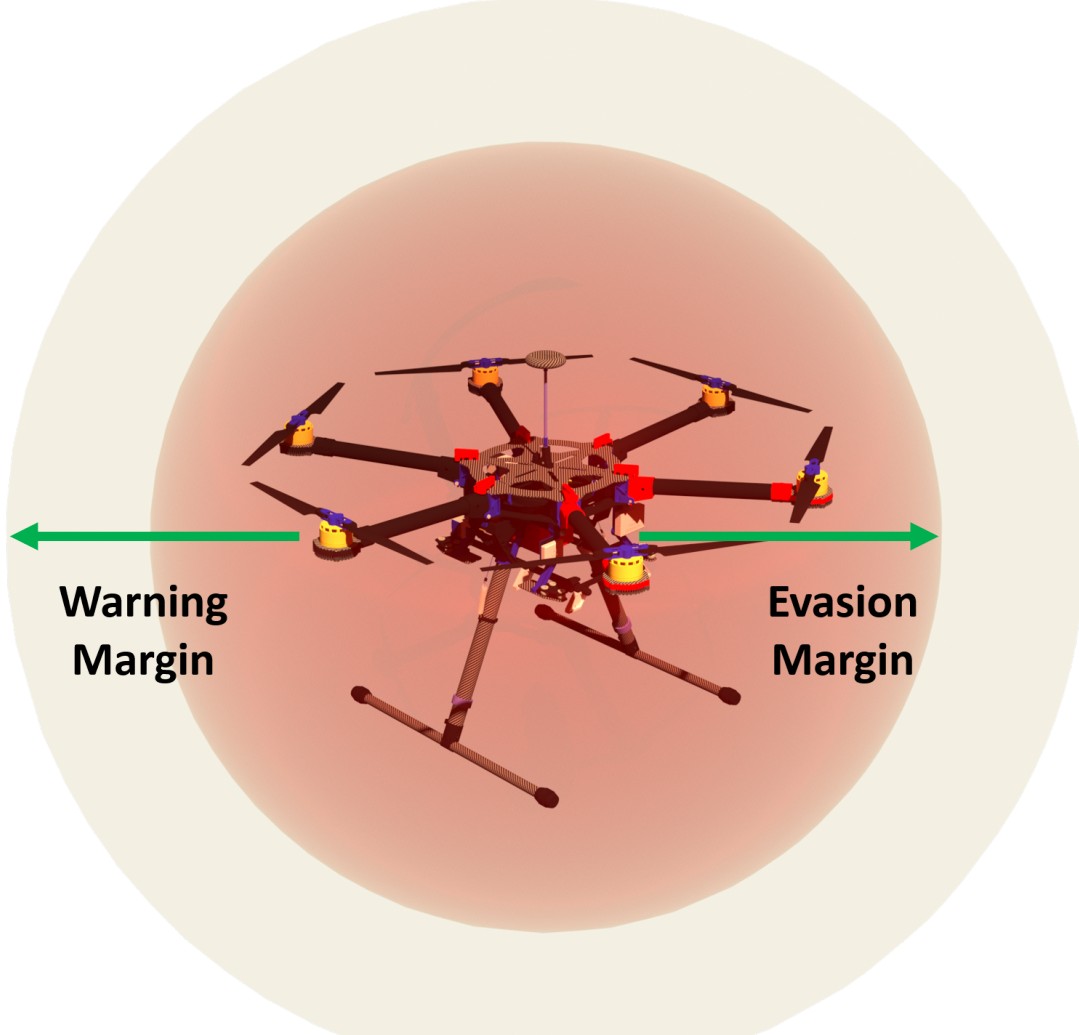

**Figure 13.** Collision avoidance system based on geofence.

The yellow outer dial is a first safety volume around the UAV whose fully parameterized radius establishes a safety margin which, if over-steer by another UAV, would issue a warning. This warning is maintained if both UAVs are within a distance between the first and second rings.

In the case of the inner sphere, red, a radius is set, again parameterized, which allows to delimit the volume from which it is necessary to perform an obstacle avoidance manoeuvre. Thus, if, despite the warning, the movement of one of the UAVs causes the distance between the two to be reduced, establishing itself below the marked operating range, an evasive manoeuvre is performed that guarantees the safety of the system. This

maneuver has been defined to resolve the conflict vertically and both UAVs modify their flight height to avoid possible collision. Once the alert situation is complete, both UAVs regain their marked altitude before the conflict is generated.

### 2.4. Layer IV: Intelligent Decision-Making System for Autonomous UAVs Navigation

In this section, a new layer, based on intelligent algorithms, is presented, and analyzed, which aims to provide the developed architecture with a module responsible for intelligent decision-making to achieve autonomous navigation by UAVs that are part of a swarm.

Advances in the field of AI have led to the expansion of these methods to a myriad of lines of research including their application to the UAVs field. The development of AI-based algorithms, and in particular the advancement in the field of Reinforcement Learning (RL), have caused the implementation of a method based on this technology to complement the architecture presented so far and generate a model that allows each UAV to navigate autonomously through the environment, thanks to the ability to make decisions for itself considering the captured information of the environment.

The development of this new layer within the architecture seeks to generate robustness and improve efficiency within it, establishing a decision-making algorithm, based on RL, that allows un supervisory navigation from one location to another through intelligent decision making, so that it is the UAV itself that establishes what action to take at every moment of time, to maximize the reward obtained along the way. In this way, a new loop of control is introduced within the architecture that guarantees the culmination of a mission in adverse environments.

RL is considered an area of machine learning characterized by data collection from a dynamic environment, with the aim of generating a sequence of actions that generate an optimal result [34]. Specifically, the RL consists of learning what to do or what situations to assign to a set of actions, to maximize a numerical signal called reward, without telling the agent what action to take, but it is himself who must discover which actions produce the greatest reward through the exploration of the environment. This feature, to learn for itself, by trial-error to determine the best behavior, is what sets this type of algorithm apart from the rest within Machine Learning (ML).

While it is true that classic RL algorithms are presented as an effective solution to low-dimensional problems, they show limitations in becoming scalable methods capable of generating solutions to high-dimensional problems [35]. To address the problem of dimensionality, a research current has emerged in recent years in charge of merging Deep Learning (DL)-based methods along withRL-based methods, which have enabled the progress of this technique in complex fields where the treatment of large amounts of information is necessary, using deep reinforcement learning (DRL) algorithms.

The use of DRL algorithms allows a step towards the creation of autonomous systems with a higher level of understanding of the environment in which they are located, causing such algorithms to be applied to decision-making problems in which the action and observation spaces have a high dimension.

The DRL presents as its main idea that the learning capacity of the agent is determined by a DL model or a Deep Neural Network (DNN). DRL algorithms typically maintain the same approaches as those used in RL algorithms with the novelty that they introduce DL models for agents to learn policies. Thus, algorithms such as Deep Q-Network (DQN) [36,37], Trusted Region Policy Optimization (TRPO) [38], and Asynchronous Actor-Critic Agents (A3C) [39] are established as the central set of methods within the DRL. The use of Convolutional Neural Networks (CNN) as a component within an agent allows you to learn policies from crude and high-dimension entries such as images [40].

The use of this method in both trajectory planning and autonomous navigation of vehicles such as UAVs is increasingly widespread mainly due to the possibility of generating a model that allows, to a UAV, a specific objective by navigating unsupervised through unknown environments, of which no prior information is available and, where through information collected from the environment and the state of the aircraft, a success-

ful decision-making can be made. In this way, thanks to the knowledge acquired by the UAV through DRL methods such as DQN, complex tasks can be undertaken in dynamic environments such as forest and urban fires. This complex learning is made possible by being able to introduce to ANNs a set of high-dimensional observations that allow the agent to learn that policy that chooses, always, that action that maximizes the return value, that is, the set of rewards when performing a sequence of actions on the environment.

Therefore, the main idea of the development proposed in this chapter focuses on being able to include, within the autonomous navigation architecture, a new layer that gives systems a sufficiently advanced AI so that each UAV chooses a sequence of actions that allows them to navigate from one path point to another avoid colliding with objects present in the environment through autonomous decision-making.

The section shows the developments performed in this area and the results achieved, comparing them based on the training parameters used, the set of states, the set of observations and the environment used.

For both algorithm training and results validation, a software architecture based on TensorFlow [41] and Keras [42] has been used integrating ROS and Gazebo, in such a way as to enable training on a simulated environment that facilitates, in the future, the move from implementation and validation to real aerial platforms.

To establish a DRL model as versatile and complete as possible that allows a UAV to navigate, from one path point to another through an unknown environment, different trainings have been carried out in which aspects such as the structure of the ANN, the space of actions or training and learning parameters have been modified.

What has remained constant throughout development are the system inputs, or what in DRL is known as observation space and which, in this case, is linked to the information coming from the UAV and, which allows to know the state of the UAV within the environment. The number of observations is 7 and corresponds to the $X$, $Y$, $Z$ position of the UAV, its orientation at *pitch*, *roll* and *yaw* angles, and finally the *distance* value to obstacles obtained by using a sonar sensor. This set of parameters constitutes the entry to the ANN and allows to know the status of the UAV at each step, being able to know if it is approaching or moving away from the objective, whether the stability of the system is correct, whether the action taken has caused the UAV to acquire a compromised physical state or, finally, to know whether it is approaching or moving away from an obstacle present in the environment.

As indicated, the observation space constitutes the input to the neural network model used during training and that, with changes in the number of layers and dimensions of the output space of each of them, presents a similar general structure in all cases, as shown in Figure 14, in the sense that the entry into the network will always be the observation space and the output of it will be characterized by the action space used at that time.

The following discusses those cases in which a model capable of navigating the environment has been obtained, without supervision, from one given location to another, reaching the previously set goal.

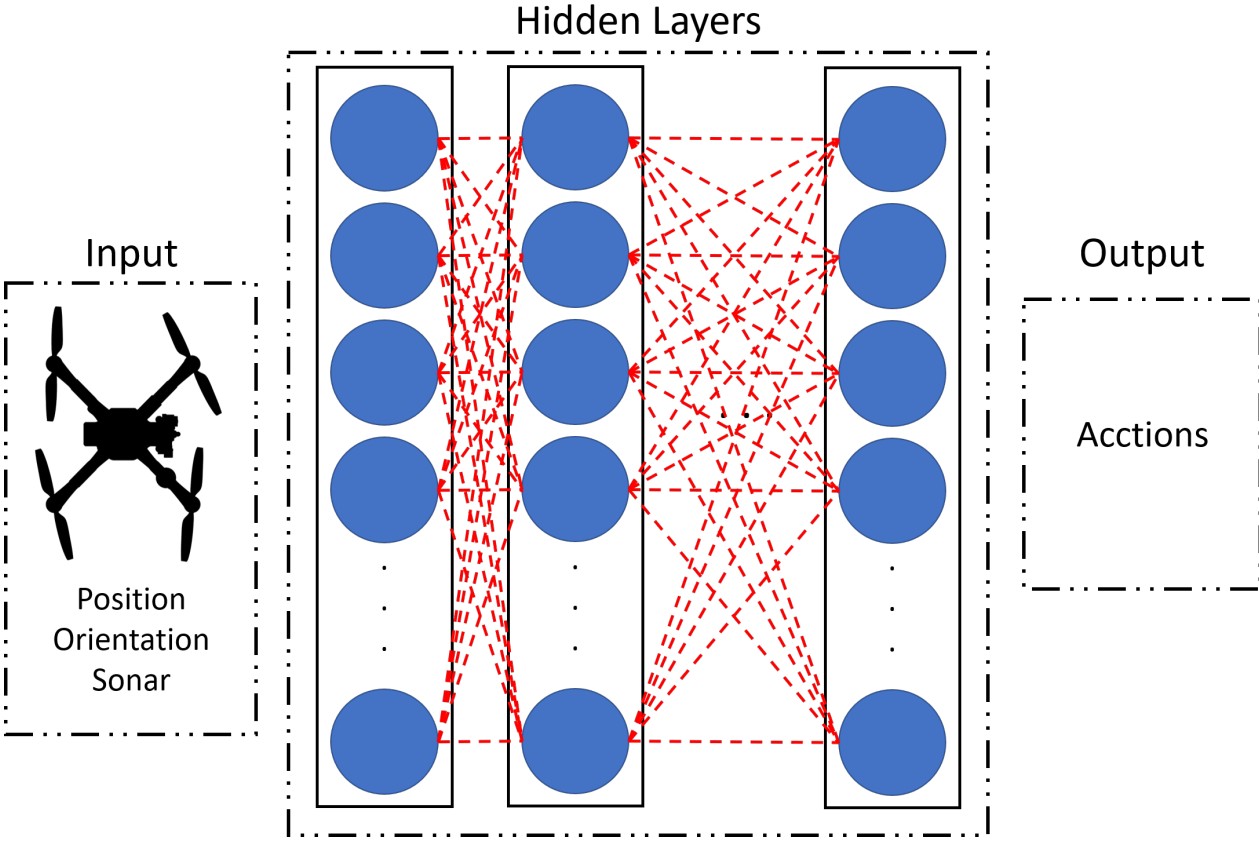

**Figure 14.** General structure of the Artificial Neural Network (ANN) used in the Deep Q-Network (DQN) algorithm.

1. **Smart Decision-Making Model 1**

    This first model generates a solution for autonomous navigation of a UAV to a particular location in a scenario with the presence of static obstacles.

    When choosing the structure of the neural network, for learning the policy to follow to maximize the reward on the part of the agent, it is important to establish a network complex enough to be able to approach the requested function, but not so complex that it makes training impossible. For this reason, and because during this development no visual information from the environment will be analyzed, which would make CNNs an ideal structure for the agent network it has been decided to begin research using the more general multilayer networks such as fully connected networks, in which each neuron in one layer is connected to all neurons in the next. In this model, a structure such as that listed in Table 3, characterized by a symmetrical structure of 5 hidden layers fully connected, has been used.

**Table 3.** Structure of the ANN for model number 1 of deep reinforcement learning (DRL).

| Layer (Type) | Output | Parameters # |
|---|---|---|
| dense_1 (Dense) | (None, 24) | 192 |
| dense_2 (Dense) | (None, 48) | 1200 |
| dense_3 (Dense) | (None, 64) | 3136 |
| dense_4 (Dense) | (None, 48) | 3120 |
| dense_5 (Dense) | (None, 24) | 1176 |
| dense_6 (Dense) | (None, 6) | 150 |

The last layer corresponds to the network output and is parameterized based on the number of actions that characterize the agent and, in this case, is limited to 6. Specifically, the agent can choose from the following set of actions—positive movement on the $X$ axis, negative movement on the $X$ axis, positive movement on the $Y$ axis, negative movement on the $Y$ axis, positive movement on the $Z$ axis, and finally negative movement on the $Z$ axis.

Within the structure of the neural network, it is necessary to use optimizers that help the convergence of these towards a local minimum. This work uses the Adaptive Moment Estimation(ADAM) optimizer [43], which combines the strengths of the Adaptive Gradient algorithm (ADA) and the Mean Quadratic Value Propagation algorithm (RMSProp) and, as a novelty, proposes the moment, or what is the same, the acceleration of gradient descent. It is a highly used optimizer today due to its low memory usage and computational effectiveness and that, it has the learning rate as an essential parameter in the operation of this. The learning rate consists of a hyperparameter that modifies network performance and should be adjusted to prevent learning from being excessively delayed, in cases where the learning rate is low, or erratic, when the learning rate is very high.

A fundamental element within the RL and DRL algorithms is the reward, that is, the numerical value that the agent receives based on how good or bad the action executed on the environment is. In this case, the following reward scheme is established: it

- **For each iteration with the environment:** After each action is executed, the algorithm checks whether the distance to the target has been reduced with that action or increased. In case the action reduces the distance to the target, a positive reward of 10is generated, while if you move away from the destination the reward is negative of value similar to the previous one. The idea of employing this reward system is to avoid the agent's oscillations in the environment, since without negative reward or, with a negative reward of less magnitude than positive, the agent tended to detect an obstacle and start performing repetitive forward and backward movements obtaining a higher accumulated reward than for cases where the UAV can navigate directly towards the target.

- **For the end of each epoch:** At the end of each era, either because the agent has reached the target, because a maximum number of iterations have been given or because there is a situation contemplated for this to happen, a reward is established that varies depending on whether the target has been reached or not and, depending on the distance to it. So, if the destination is reached, a reward is generated that follows Equation (1), while if the target is not reached the generated reward follows Equation (2). This rewards you, whether the target has been achieved or not, proximity to the destination location. This solution allows the agent to have an understanding of how good this sequence of actions has been with respect to the previous ones, even if the objective is not achieved, since, by entering the parameter of the distance to the objective, even if the training is not satisfactory at that time, the agent can know if it is better than other sequences where the path point in question has not been reached and, in this way, improve their knowledge and learning.

$$\gamma = 200 - 0.1 \cdot d \tag{1}$$

$$\gamma = -200 - 0.1 \cdot d, \tag{2}$$

where $\gamma$ corresponds to the reward value and $d$ to the distance to the target.

To improve learning and accelerate the convergence of training, several cases have been established for which the time is over, the environment is restarted and started again. These cases are determined by the observation space and the interpretation of these by the algorithm.

Thus, the time is terminated if the UAV has collided with an obstacle or is in a situation incompatible with the flight stability of the system. To do this, the value of the sensor is first checked to sound on board, so that if that value is below 0.5 m it is assumed that the aircraft will collide with an obstacle of the environment. Secondly, pitch and roll angles are checked and, if they are above 90°, the UAV is considered to have reached a position incompatible with the stability and flight of the system. Finally, to speed up the learning process, during training a work space is established that delimits an area through which the agent can be moved, ending the epoch, and starting a new one in the event that an action leads the UAV to a location outside the bounded area. As shown in Figure 15, as the training progresses, it can be seen how the learning of the network improves and the decision-making model generated approaches a policy that manages to maximize the value returned, showing that the structure of DQN used in this model is satisfactory.

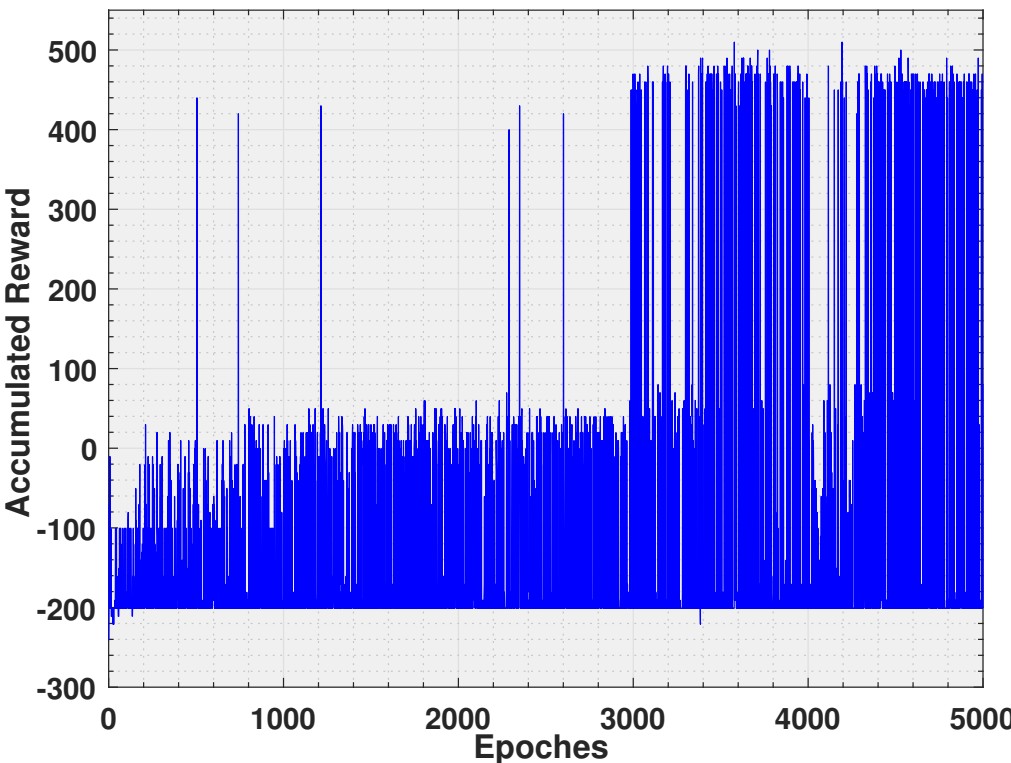

**Figure 15.** Accumulated reward per training epoch for decision making model 1.

Figure 15 shows the total reward obtained at the end of each training season, and it shows the essence of DRL algorithms based on trial-error learning. In the early days, when the scanning rate is higher, the agent establishes sequences of actions that are not satisfactory by accumulating, at the end of each era, a negative total reward. As training progresses and, the agent first learns what are the best actions to take and, secondly, the exploration rate is reduced to prioritize the exploitation of the best actions, it can be observed how the reward accumulated at the end of each era is being more positive, increasing with training and achieving some stability around the maximum reward value at the end of the training.

This analysis is completed with Figure 16, which shows the total reward accumulated as training progresses. In it, you can see how at the beginning of the training, the lack of knowledge and learning and, the configuration of the training to perform a high exploration at the beginning in search of the sequence of actions that maximizes the returned value causes the accumulated reward to decay towards negative values in the first half of the training. As the agent gains learning, the returned value improves

and increases, and the abandonment of exploration to the detriment of exploration causes the slope of the curve to soften the accumulated reward. The result of the curve makes it possible to conclude that, although the result obtained is satisfactory in terms of autonomous navigation of the UAV, a larger set of epochs would be necessary to reverse the slope of the curve, establish a growth of the accumulated total reward and end up stabilizing that curve in positive reward values.

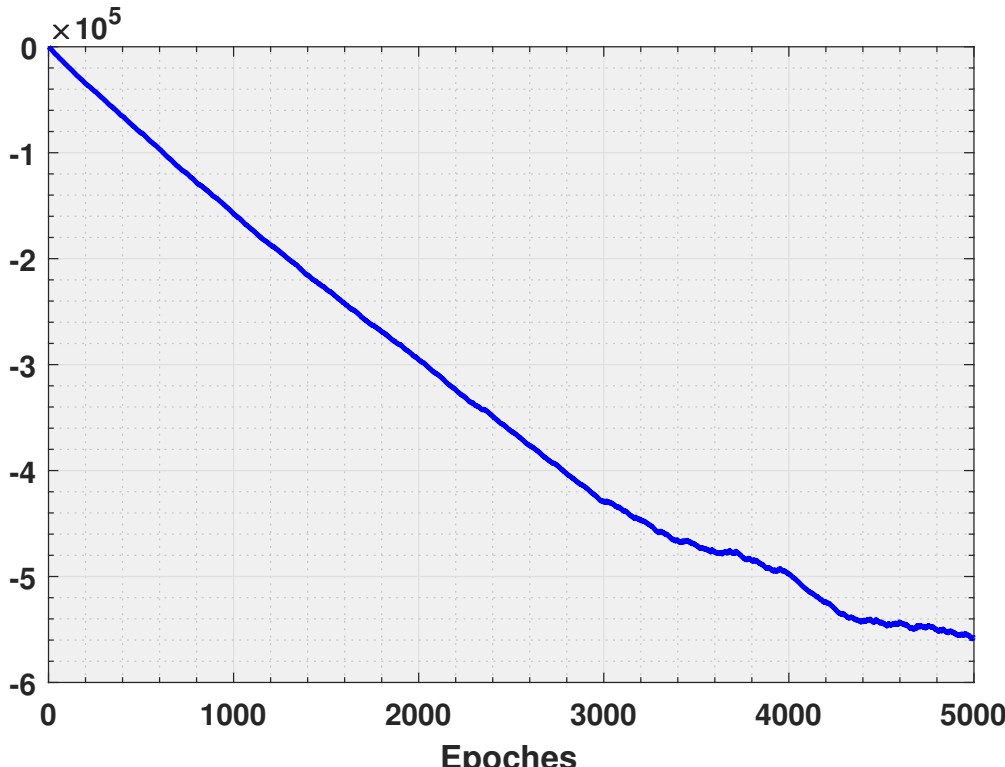

**Figure 16.** Reward accumulated throughout the training for the intelligent decision-making model 1.

It is important to note that the DQN algorithm is characterized by employing a repeating approach to experience to eliminate problems related to correlations between consecutive samples. For this reason, this model is configured so that a maximum of 64 batches can be stored in memory, allowing the Q values to be updated from a random sampling of that repeat memory.

Next to this, another set of parameters are established that seek to address the problem of exploration-exploitation. Therefore, a variable is included that allows to control the exploration, so that at the beginning of the training favors it, but that this constant decays as the times increase, thus favoring exploitation as the training progresses.

From this training, you get a model that allows a UAV to navigate, without supervision, to a destination location over an environment like the one explored.

2. **Smart Decision-Making Model 2**

This second model generates a solution for autonomous navigation of a UAV to a particular location in a scenario with the presence of static obstacles identical to that chosen for the first learning model. The goal of this second model is to continue to delve deeper into DRL algorithms and try to give the UAV more similar behavior to real flight.

Since, in the previous case, the structure of the neural network achieved satisfactory results and since the observation space remains immovable, this second model has been chosen to use a multi-layer network-based structure as in the previous case. For this model, a symmetrical network has been opted again, as shown in Table 4,

fully connected, but modifying the structure of each of the hidden layers, varying the number of neurons because, in this case, a different action space is used, with which the previous neural network structure was not able to obtain correct learning.

**Table 4.** ANN structure for the DRL decision-making model number 2.

| Layer (Type) | Output | Parameters # |
|:---:|:---:|:---:|
| dense_1 (Dense) | (None, 48) | 384 |
| dense_2 (Dense) | (None, 64) | 3136 |
| dense_3 (Dense) | (None, 128) | 8320 |
| dense_4 (Dense) | (None, 64) | 8256 |
| dense_5 (Dense) | (None, 48) | 3120 |
| dense_6 (Dense) | (None, 6) | 294 |

Again, the last layer corresponds to the network output and is parameterized based on the number of actions that characterize the agent and, in this case, is limited to 6. Specifically, for this second model and, in order to obtain a more real behavior, the agent can choose between the following set of actions: positive movement on the $X$ axis, negative movement on the $X$ axis, positive rotation on the $Z$ axis, negative rotation on the $Z$ axis, positive movement on the $Z$ axis and finally negative movement on the $Z$ axis. In this way, with this new action space, the UAV can perform ascent and descent movements, go forward or backward or, finally, make turns along the Z axis or yaw, allowing you to establish a control, not only of position, but of orientation of the vehicle.

As for rewards, it is important to make changes from the previous model to consider new aspects introduced such as orientation. For this reason, the following reward scheme has been established in this model:

(a) **For each iteration with the environment:** After each action, the implemented algorithm is responsible for checking both the new distance to the target and the new orientation reached by the UAV. This orientation compares to the previous orientation and target orientation, so that if the agent has set a rotation action on the Z axis, it can have feedback on how good or bad such action has been. The target orientation is considered to the $\alpha$ angle that the UAV should rotate from its current orientation to align its $X$ axis with the target, as shown in Figure 17.

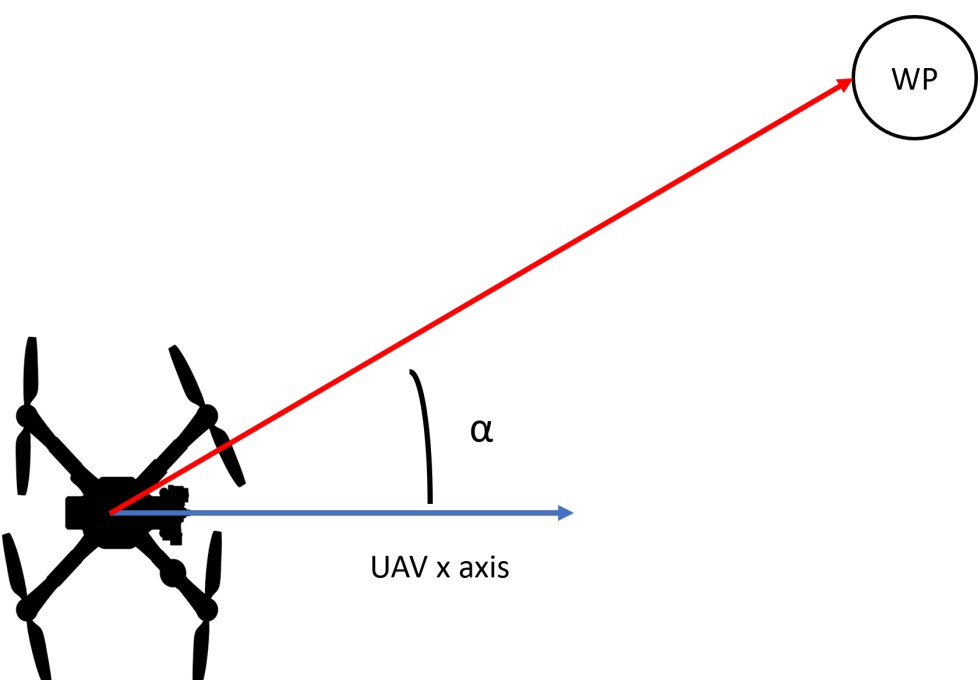

**Figure 17.** Orientation calculation scheme to allow autonomous navigation with the UAV focused on the target to be reached.

With these considerations, the reward structure for each action is as follows:

- If the distance to the target decreases and the orientation angle approaches the desired one, a reward of value 20 is obtained.
- If the distance decreases, but the UAV rotation has taken you away from the desired orientation, the reward value is set to 10.
- If the distance value to the target increases, the earned reward is set to a value of −30. This value is set by looking for agent oscillations around a cumulative location reward without performing an optimal sequence of actions.

(b) **For the end of each epoch:** At the end of each era, the reward structure used in the previous model is followed, which varies depending on the distance to the target, being positive if the same or negative has been achieved otherwise.

In this case, Figure 18 collects the reward accumulated at the end of each era throughout the training. It can be seen how in this case, faster convergence occurs, reducing the number of epochs used during training. This is due to two reasons, the first, the training is configured so that, if the average of the accumulated reward is above a threshold the training is terminated, considering that the model has acquired the necessary learning to achieve the goal. The second, because in this case a pre-trained model whose results were not completely valid has been used but possessed an apprenticeship that approximated the desired behavior.

Again, if you look at Figure 18 it is observed that at the beginning of the training the oscillations are higher because the scanning rate is higher and, although in this case the model starts from a previous learning, sometimes the excess exploration leads to a sequence of actions in which the result is not satisfactory, obtaining a negative return value. As training increases, the decrease in exploration in favor of further exploitation causes the agent to chain times with satisfactory results, more consistently obtaining a maximum return value. However, the scan is not reduced to 0, so even at the end of the workout there are times when the sequence of actions set is not correct.

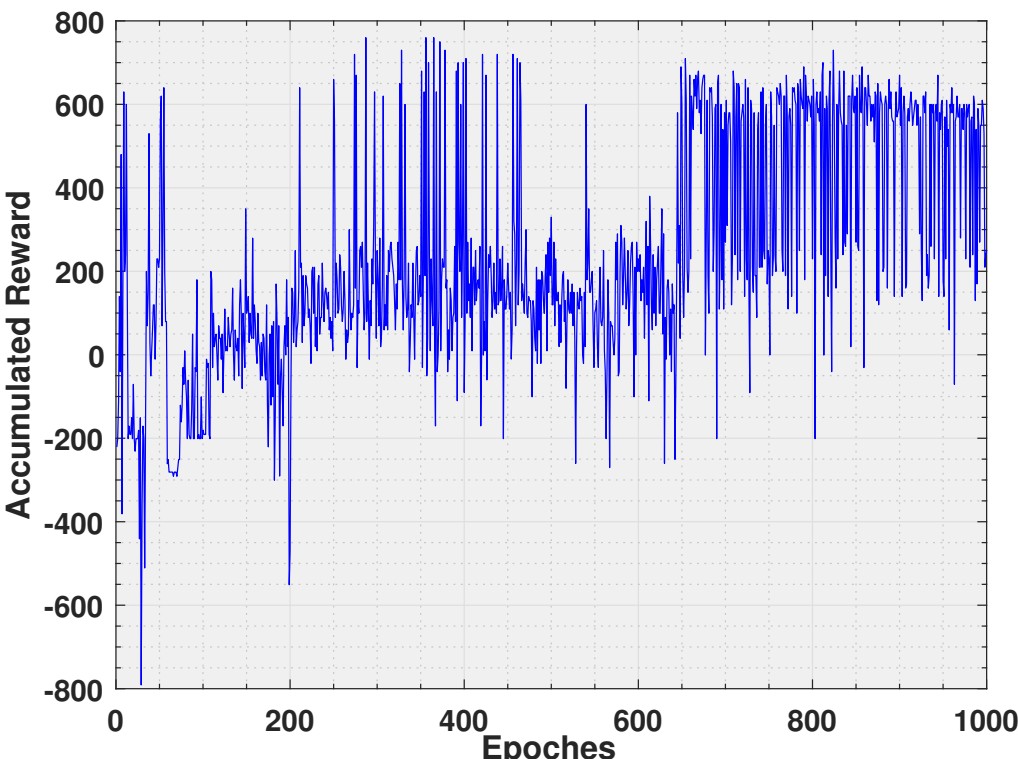

**Figure 18.** Accumulated rewards for Model 2 decision making.

As in the previous model, the development and implementation analysis performed is completed with Figure 19. It shows the reward accumulated throughout the training and, in this case yes, it can be observed how the profile of the graph adapts to the expected optimal result of the DRL algorithms, that is, it begins with a negative cumulative reward resulting from the lack of learning and the exploration phase and, as the training progresses, the model acquires greater knowledge, abandons exploration to the detriment of exploitation and the accumulated reward gradually grows to be at maximum return values.

The idea of using these previously trained models is to reduce, as shown in Figure 18, the time it takes for the network to converge towards an optimal policy in which the accumulated reward is maximized. Based on the network structure set out in the previous model, the algorithm loads the training parameters and begins the iterative process trying to achieve an optimal policy that allows the agent to successfully complete the marked goal.

If you analyze the results of the pre-trained model, it can be observed as modifying the ANN structure by increasing the number of fully connected nodes in each of the layers allows for faster convergence in the learning phase. In this case, the model manages to chain times with a reward accumulated by high time, generating a model that is able to navigate to the pre-set area of interest, but with improved accuracy, hence this model was used to perform a new training and, try to improve the accuracy of this. Add up the two training processes, it is established that 1749 epochs are required to obtain a model capable of navigating between two specific locations completely autonomously.

The previously trained model was capable of unsupervised navigation, but the shortest distance to the destination stood at 1.98 m, while for this model in question an approximation of the location of interest of less than 0.8 m is achieved.

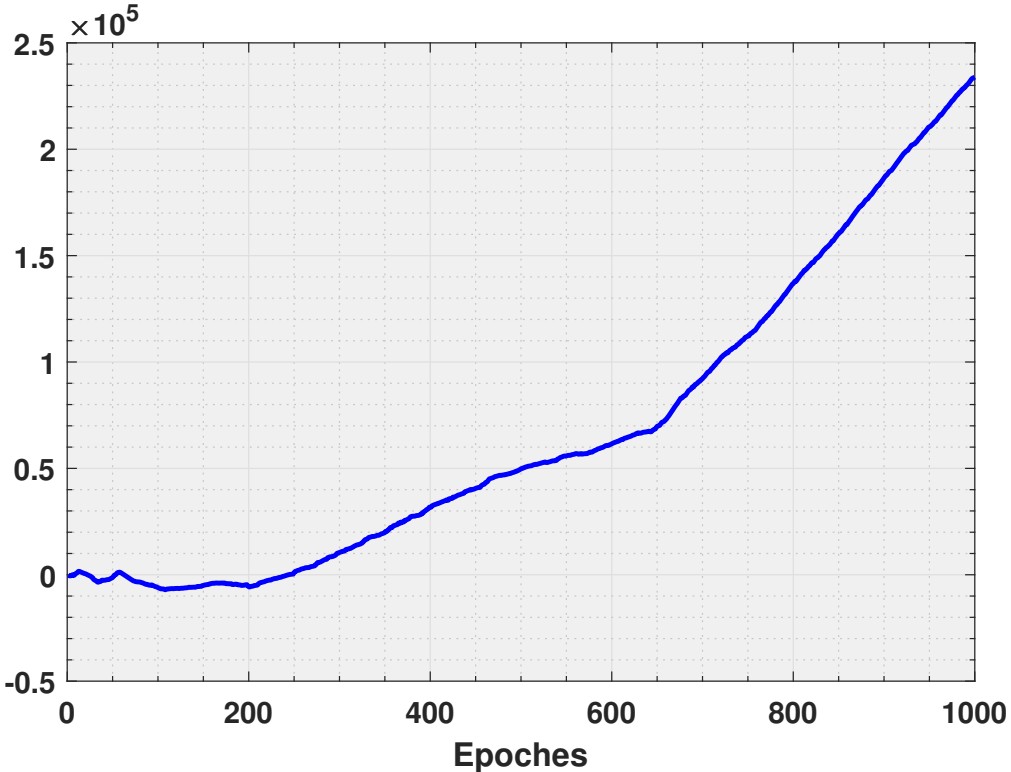

**Figure 19.** Reward accumulated throughout the training for the decision-making model 2.

When comparing this learning model, in which the action state has been configured to generate more realistic behavior from the point of view of stability and flight dynamics, with the learning model 1, it can be observed that, for this new model, the agent is able to achieve the target with greater precision for a similar total travel distance. Thus, the smart decision-making model 1 travels 11.71 m navigating the environment to reach the target with an accuracy of 0.99 m, while model 2 can reach the destination with an accuracy of 0.70 m after having traveled 11.85 m.

3.　**Smart decision-making model 3**

This third model generates a new solution for autonomous navigation of a UAV between two specific locations in a scenario with the presence of static obstacles, similar to that used in previous learning models. The goal of this third model is to continue to delve into DRL algorithms and try to improve accuracy when reaching a particular location by introducing restrictions when performing actions during training.

To improve the accuracy with which the UAV navigates to a preset location, a triangle-shaped delimited work space has been established, with a vertex at that location, so that the closer it gets to the target, the less moving capacity the *XY* plane the agent has. To better detail and understand this UAV mobility restriction as it approaches the target, Figure 20 has been included, which shows an outline of how dimensioning is performed on the *XY* plane of the available work space.

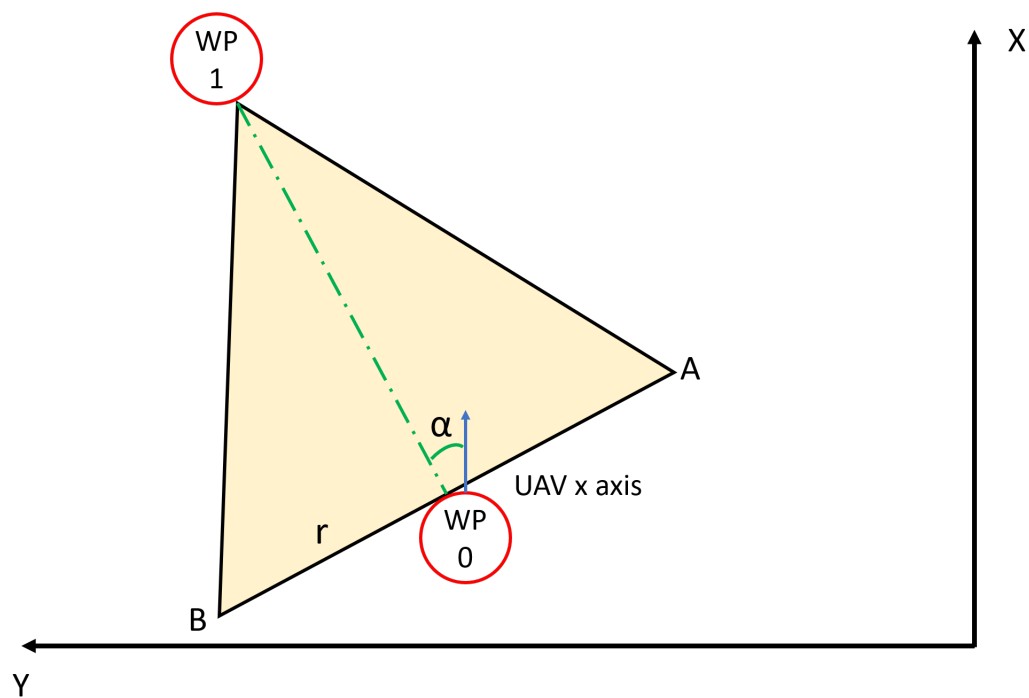

**Figure 20.** Limiting the scan area for precise navigation to a preset location.

This way, depending on an $r$ radius, a triangle is established on the $XY$ plane on which the UAV can be moved in search of an optimal path to the set target ($WP_1$). This space (area shaded in yellow) has been dimensioned along the $XY$ plane and has not been extrapolated to the third dimension, such as a cone-shaped volume, to allow the agent to dodge obstacles through up and down movements, if there was not enough space on the $XY$ plane to perform avoidance maneuvers.

For its implementation within the DRL algorithm, after each action is performed, a verification process is established to establish whether the agent continues to move within the space chosen for it or if, on the contrary, the action taken has led it to leave it, which is why the environment will be restarted, finally at the end of the time, accumulating the corresponding reward and starting a new epoch.

From a development point of view, a method is implemented to generate a regular polygon with vertex at the target point, point A, and point B, and at each iteration iteration is taken, from the state space, the current position of the agent, and checks whether that location is inside or outside the triangle. Trigonometry concepts are used for the calculation of points A and B and, from the $\alpha$ angle, the starting point of the UAV ($WP_0$) and the preset radius ($r$) the $X$ and $Y$ coordinates are calculated using Equations (3) and (4).

$$X_A = -X_B = X_{WP_0} + r \cdot \sin(\alpha) \tag{3}$$

$$Y_A = -Y_B = -(Y_{WP_0} + r \cdot \cos(\alpha)). \tag{4}$$

This third model maintains the ANN structure that allows for satisfactory results in previous models, using fully connected multilayer networks. For this model, a symmetrical network has been opted again, as shown in Table 5, but by modifying the structure by removing one of the hidden layers, trying to check whether for this case, a network with fewer connected nodes is able to learn behavior such as that intended for the agent.

**Table 5.** Structure of the ANN for DRL Model 3.

| Layer (Type) | Output | Parameters # |
| --- | --- | --- |
| dense_1 (Dense) | (None, 48) | 384 |
| dense_2 (Dense) | (None, 64) | 3136 |
| dense_3 (Dense) | (None, 64) | 4160 |
| dense_4 (Dense) | (None, 48) | 3120 |
| dense_5 (Dense) | (None, 6) | 294 |

As for rewards, since the only variation from the previous learning model is the modification of the work space to achieve greater accuracy when approaching a given location, it has been chosen to maintain the same reward system as for smart decision-making model 2.

Figure 21 shows the evolution of the reward accumulated at the end of each era by the agent throughout the training. In this case, a previously trained model has been used again, which had not achieved a completely satisfactory result at the end of the training, but whose generated learning model approximated the desired behavior. It can be seen how, starting from a pre-trained model, only 1925 eras are necessary to achieve the convergence of the ANN towards an optimal policy that maximizes the total reward accumulated by the agent.

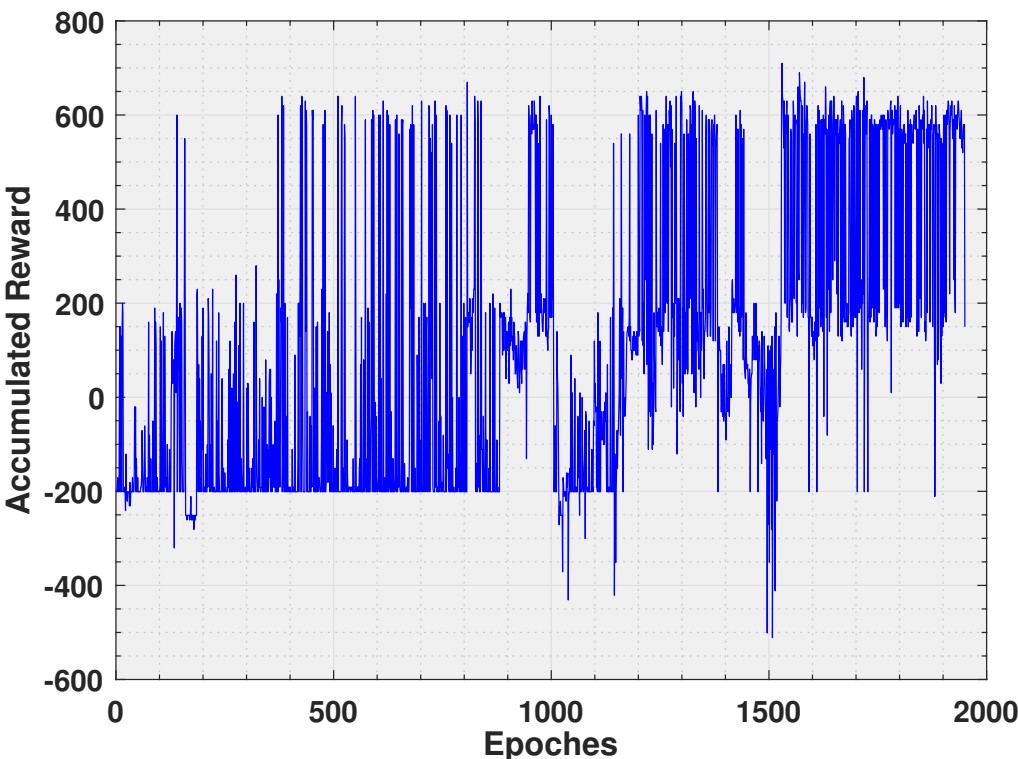

**Figure 21.** Reward accumulated in each epoch by the model 3 of intelligent decision making.

In this case, Figure 21 includes the results of the previously trained model hence, at the beginning of the entire graph the results obtained are negative, the training improves and, in the middle of the training negative values reappear. These negative spaces that appear throughout the 1000 and 1500 eras are because the training of the model begins from previous learning, so the exploration phase is again high and, despite having previous learning, periods are interspersed with a sequence of

successful actions, with exploration times where the result moves away from the goal to be achieved.

Alongside this analysis, the advancement of the total reward acquired throughout the training is included, as reflected in Figure 22. Again, it can be observed how the response obtained approximates the expected result and, is that after a set of times in which the agent explores and try to find new optimal alternatives, finally reaches the phase of exploitation in which the agent chooses those actions that returns a positive reward value and, the return value accumulates obtaining maximums towards the end of the training.

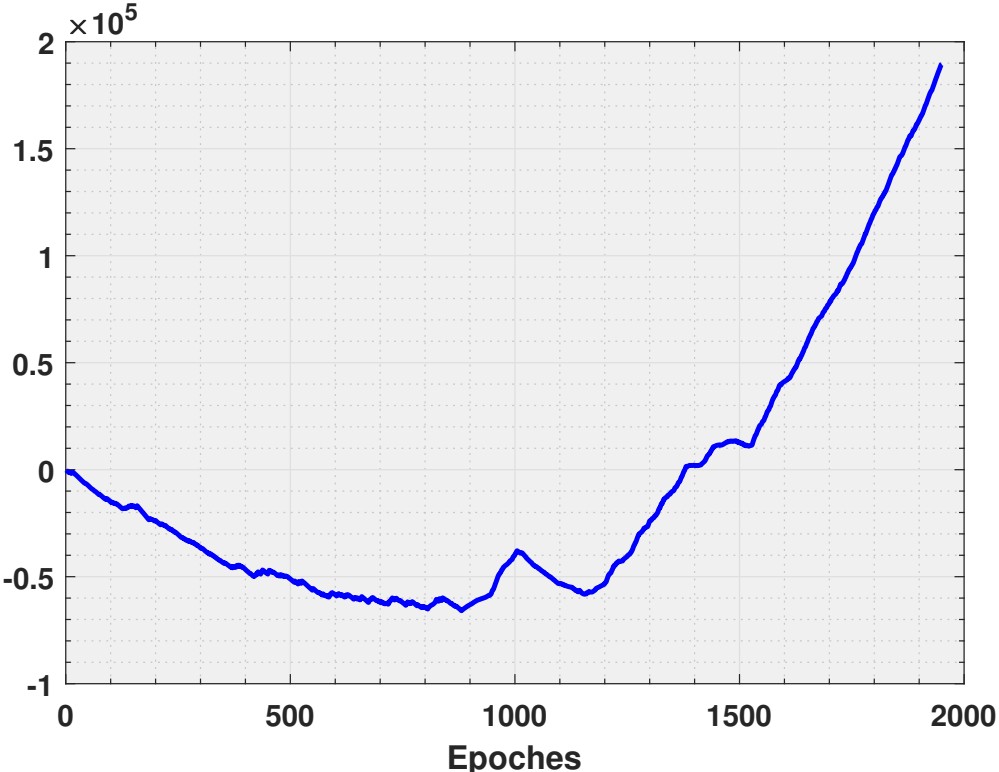

**Figure 22.** Cumulative total reward for Model 3 intelligent decision making over the course of the training.

Again, it is important to establish a comparison on how the changes made in terms of the movement restriction of the UAV within the environment, cause or not, a considerable improvement in the accuracy with which the target point is reached. Table 6 shows a comparison of the use of models generated over the same environment to reach the same destination. This table shows, on the one hand, the distance travelled in each action and how the distance to the target intended to be reached varies. If you look at the last row of each model, it can be observed as for all cases the total distance travelled is very similar, oscillating between 11.71 m and 11.85 m, while, in the case of accuracy, it is the last model, thanks to the designed work space, which achieves a 34% reduction in precision reduction, setting that value at 0.65 m.

Therefore, this learning model allows to obtain a solution for autonomous navigation of a UAV improving the accuracy with which the target locations are reached thanks to a limitation of movements in the plane *XY*, which restricts the free area as the agent approaches the target.

<p style="text-align:center"><strong>Table 6.</strong> Comparison between learning models.</p>

| Learning Model 1 | | Learning Model 2 | | Learning Model 3 | |
|---|---|---|---|---|---|
| **Travelled Distance** | **Distance to Target** | **Travelled Distance** | **Distance to Target** | **Travelled Distance** | **Distance to Target** |
| 0.66 | 9.91 | 1.00 | 9.92 | 1.00 | 9.92 |
| 1.07 | 9.58 | 1.37 | 9.76 | 1.36 | 9.76 |
| 1.55 | 9.16 | 1.82 | 9.37 | 1.81 | 9.37 |
| 2.04 | 8.74 | 2.32 | 8.94 | 2.31 | 8.94 |
| 2.54 | 8.33 | 2.82 | 8.52 | 2.81 | 8.53 |
| 3.04 | 7.92 | 3.31 | 8.12 | 3.31 | 8.12 |
| 3.46 | 7.68 | 3.82 | 7.70 | 3.81 | 7.71 |
| 3.78 | 7.47 | 4.32 | 7.29 | 4.31 | 7.30 |
| 4.25 | 7.09 | 4.82 | 6.90 | 4.81 | 6.91 |
| 4.75 | 6.71 | 5.32 | 6.52 | 5.32 | 6.52 |
| 5.25 | 6.33 | 5.82 | 6.15 | 5.60 | 6.31 |
| 5.75 | 5.98 | 6.11 | 5.96 | 5.86 | 6.10 |
| 6.26 | 5.67 | 6.18 | 5.93 | 6.33 | 5.70 |
| 6.62 | 5.32 | 6.39 | 5.73 | 6.61 | 5.47 |
| 7.08 | 4.92 | 6.86 | 5.31 | 6.65 | 5.43 |
| 7.44 | 4.56 | 7.15 | 5.05 | 6.86 | 5.22 |
| 7.91 | 4.30 | 7.40 | 4.80 | 7.32 | 4.78 |
| 8.27 | 3.98 | 7.87 | 4.34 | 7.82 | 4.28 |
| 8.73 | 3.56 | 8.37 | 3.85 | 8.32 | 3.79 |
| 9.08 | 3.21 | 8.86 | 3.36 | 8.82 | 3.29 |
| 9.54 | 2.99 | 9.37 | 2.87 | 9.32 | 2.80 |
| 9.90 | 2.71 | 9.87 | 2.39 | 9.82 | 2.31 |
| 10.37 | 2.24 | 10.37 | 1.92 | 10.32 | 1.83 |
| 10.87 | 1.78 | 10.87 | 1.47 | 10.83 | 1.37 |
| 11.36 | 1.33 | 11.37 | 1.06 | 11.33 | 0.94 |
| **11.71** | 0.99 | 11.85 | 0.78 | 11.83 | **0.65** |

## 4. Smart Decision-Making Model 4

So far, all the learning models analyzed have worked on a static environment, with a low density of obstacles, but have allowed to generate dynamic models of DRL-based UAVs capable of introducing a stand-alone decision-making layer within the presented architecture that increases the robustness of the system to navigate to a specific location, without supervision.

To be able to know how DRL-based developments allow to introduce into the architecture a new layer capable of complementing the planner described in the first layer, a model has been established capable of autonomously navigating a structured environment with the presence of dynamic obstacles that corresponds to one of the environments used for the validation of the methods of path planning and autonomous navigation, to which the presence of dynamic obstacles has been added. The idea of including such obstacles within the environment is to be able to establish a model that has the ability to perform tasks in changing environments such as fires, being able to generate correct responses and decision-making in case of detecting the presence of any obstacle. In this way, the agent can make decisions in the presence of external objects, such as other UAVs in the environment or obstacles present in the work space.

For this fourth model, a structure of fully connected hidden layers has been maintained, allowing to generate a high density of knowledge based on a highly connected network. Specifically, the structure used is like that set out in learning model 2, as shown in Table 7 of the neurons in each of the layers, since, although the observation space remains constant, the information coming from the environment implies greater complexity in the presence of dynamic obstacles.

**Table 7.** Structure of the ANN for model number 4 of DRL.

| Layer (Type) | Output | Parameters # |
|---|---|---|
| dense_1 (Dense) | (None, 48) | 384 |
| dense_2 (Dense) | (None, 64) | 3136 |
| dense_3 (Dense) | (None, 128) | 8320 |
| dense_4 (Dense) | (None, 64) | 8256 |
| dense_5 (Dense) | (None, 48) | 3120 |
| dense_6 (Dense) | (None, 6) | 294 |

As for the other fundamental elements of DRL algorithms, everything remains constant with respect to the last model implemented, both in terms of the state of actions and the structure of the rewards.

As detailed throughout this work, in jobs where response time is a key aspect of mission success, the distance traveled by UAVs is an essential parameter in assessing the proposed methods, since, in the face of the use of similar systems, where dynamic aspects such as speed are identical, the time spent achieving a goal is determined by the distance travelled to that location. For this reason, the proposed decision-making model has been tested and validated on a scenario used for validation of 3D planning methods, checking whether the DQN algorithm used allows each UAV to be sufficiently intelligent to provide at least one solution equally efficient as the previous methods proposed.

If you analyze the behavior of the learning model for this case, in which you intend to reach a final location, you can observe, by means of the data received from the positioning of the UAV, how as the agent navigates unsupervised through the environment is able to reduce the distance to the destination, being able to reach it, within a tolerance of less than 0.5 m, after having traveled 22.74 m. As for the proposed 3D path planning algorithm, for the same environment, starting from the same starting position and trying to reach the same final location, a safe, collision-free trajectory is established, the total distance of which is 24.05 m, with this comparison summarized in the Table 8.

**Table 8.** Path optimization proposed by the 3D planner through the use of DRL algorithms.

| | DQN-Based Path | Initial Path |
|---|---|---|
| **Travelled distance (m)** | **22.74** m | 24.05 m |

Therefore, because of this case study, one can see how this layer of architecture allows the swarm to be equipped with some intelligence to undertake a sequence of actions that allows them to reach a destination location minimizing the distance traveled by the UAV. In this way, this development layer could act on the control of each of the agents to establish the maneuvers to be performed to navigate unsupervised through paths generated by the proposed planning method. Alongside this, this intelligent decision-making method is positioned as a complementary tool to the obstacle detection and avoidance systems presented in the previous layer, being able to exercise as a third

control loop and assist in the optimization of avoidance maneuvers to be performed in case of detecting an obstacle on the previously fixed path.

Currently, it could be used in environments in which a previous training has been performed, being able to generate a new control loop, which is compared with the response given by the obstacle detection and avoidance layer and be able to discriminate which of the two actions is better. In this case it would act as an extra layer, giving robustness to the system, to establish a redundant architecture like that of the autopilots of traditional commercial aircraft. Thus, in the presence of an obstacle, the system would generate on the one hand a speed control with a specific module and direction, and on the other hand, the artificial intelligence algorithm would generate an action to avoid it, both would be compared, being able on the one hand to verify that both responses are similar or, if they differ, to establish a decision making system that values both inputs and establishes which one is better. Even so, as stated in Section 3, future work is oriented to increase the training environments and improve the DRL algorithms to achieve a versatile model, with a high adaptability to different environments, and that allows each UAV to navigate autonomously through unknown areas, so that this model acts as the main system for autonomous navigation, replacing this layer and only considering the exit of this in cases of emergency.

## 3. Conclusions and Future Work

This work presents a software architecture in which a set of methods are combined to allow autonomous and coordinated navigation of a swarm of UAVs in environments of various kinds, such as fires declared in forest or urban environments. The main contribution of this work is to develop a software architecture consisting of multiple layers in which, through the implementation of a set of methods and algorithms we seek to generate solutions that allow the autonomous and coordinated navigation of a swarm of UAVs in order to be able to carry out a joint mission. The proposed solutions seek to establish a robust, scalable, and secure navigation system for UAVs groups that solves problems such as safe and effective path planning, obstacle detection and avoidance, coordinated navigation under a given training, or autonomous decision-making based on captured environment training.

The global centralized architecture allows to have a greater number of computational resources in the central node located on the ground, in order to execute those methods that require a greater computational expenditure, leaving the embedded computer on board the UAV for the decentralized layer, which is formed by a set of algorithms whose implementation can be supported by embedded computers such as the Jetson AGX Xavier [44], which has been chosen to be embarked on real aerial platforms with which to begin to validate and test the proposed architecture.

Throughout this work, the potential application of UAV swarms has been demonstrated to act autonomously and coordinately different environments, through the implementation of path planning and autonomous navigation algorithms. The algorithms and approaches proposed in this work have shown different solutions, with high levels of robustness and precision, in autonomous and coordinated navigation applications of UAVs.

However, to increase the performance of these algorithms, different areas will be investigated in future work. Such future research focuses on the following areas:

- In the development and construction of a real swarm of UAVs, with structure like that presented throughout this work, to be used as a demonstrator of the algorithms and methods implemented.
- One of the crucial aspects that limits the use of UAVs is the ability to charge and the autonomy of systems. For this reason, it is essential to be able to count software tools that allow to obtain the most information of the environment from the use of low weight and consumption systems. Currently, work is under way on the development of DL-based methods that allow the generation of three-dimensional occupancy maps from the visual information obtained by a monocular camera. The output of

this method may be used as input to the methods of re-planning paths for obstacle avoidance proposed in this work.

- One of the problems detected in the literature in DRL methods is the lack of standardization of the models and the need to adapt the workouts to the environments in which you want to work. Although advances and research in the field of DL are offering solutions to this issue. For this reason, consideration is given to including more complex ANNs to include visual information from sensors on board the UAV in the observation state. In this way, increase and improve the learning of the agent to achieve a robust, heterogeneous and standard model capable of working autonomously on various types of environments. To achieve this objective, it will be advisable to carry out training of the agent in a wider range of environments, in which the complexity is increasing. In addition, increasing input information to the agent will not only bring changes in the structural complexity of the networks used, but will involve the use of more complex DRL algorithms.

**Author Contributions:** Conceptualization, Á.M. and A.A.-K.; Investigation, Á.M., A.A.-K. and D.M.; Methodology, Á.M.; Resources, P.F.; Software, Á.M.; Supervision, P.F., D.M. and A.d.l.E.; Writing—original draft, Á.M. and A.A.-K.; Writing—eview & editing, D.M. and A.d.l.E. All authors have read and agreed to the published version of the manuscript.

**Funding:** This work was supported by the Comunidad de Madrid Government through the Industrial Doctorates Grants (GRANT IND2017/TIC-7834).

**Institutional Review Board Statement:** Not applicable.

**Informed Consent Statement:** Not applicable.

**Data Availability Statement:** Not applicable.

**Acknowledgments:** The authors would like to express their gratitude to Drone Hopper (Spanish heavy-duty commercial drones Company) for providing comments on different aspects of this work and financial support. We also gratefully acknowledge the support of NVIDIA Corporation with the donation of the GPUs used for this research.

**Conflicts of Interest:** The authors declare no conflict of interest.

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
