# Peer review of "Software Architecture for Autonomous and Coordinated Navigation of UAV Swarms in Forest and Urban Firefighting"

_applsci, doi:10.3390/app11031258_

Round 1

Reviewer 1 Report

This paper addresses software architecture for autonomous navigation of UAV swarm. This is a big paper that includes many issues. As a result, the length of paper is too long. 

One concern is embedded software. How can the authors implement all software algorithm including trajectory planning, smoothing and decision making algorithms? It may requires very expensive computation power. Do you intend to use cloud for computation? How can you distribute the computation? That is my major concern. 

Reviewer 2 Report

This is an interesting paper with a practice application scenario. I have the following comments:

- the authors need to revise the abstract completely. Currently it has too many background information, and little technical information about the paper. You should summarise the challenges of the application, your approach and novelties and a brief summary of your many results. 

-Regarding the layers 3 and 4, it is not clear when the system uses the centralized, decentralized or RL methods for avoiding obstacles.
- Related to the previous point, what happens if one UAV loses the connection with the swarm, is the architecture able to maintain a centralized and decentralized configuration at the same time. This point needs to be clarified.
- The RL based method has been trained and validated with two open environments, it is recommended to work with more complex environments.
